# DEPTS: Deep Expansion Learning for Periodic Time Series Forecasting

**Wei Fan**[1]*, **Shun Zheng**[2], **Xiaohan Yi**[2], **Wei Cao**[2], **Yanjie Fu**[1], **Jiang Bian**[2], **Tie-Yan Liu**[2]
[1]University of Central Florida  [2]Microsoft Research
weifan@knights.ucf.edu, yanjie.fu@ucf.edu,
{shun.zheng, xiaohan.yi, wei.cao, jiang.bian, tyliu}@microsoft.com

## Abstract

Periodic time series (PTS) forecasting plays a crucial role in a variety of industries to foster critical tasks, such as early warning, pre-planning, resource scheduling, etc. However, the complicated dependencies of the PTS signal on its inherent periodicity as well as the sophisticated composition of various periods hinder the performance of PTS forecasting. In this paper, we introduce a deep expansion learning framework, DEPTS, for PTS forecasting. DEPTS starts with a decoupled formulation by introducing the periodic state as a hidden variable, which stimulates us to make two dedicated modules to tackle the aforementioned two challenges. First, we develop an expansion module on top of residual learning to perform a layer-by-layer expansion of those complicated dependencies. Second, we introduce a periodicity module with a parameterized periodic function that holds sufficient capacity to capture diversified periods. Moreover, our two customized modules also have certain interpretable capabilities, such as attributing the forecasts to either local momenta or global periodicity and characterizing certain core periodic properties, e.g., amplitudes and frequencies. Extensive experiments on both synthetic data and real-world data demonstrate the effectiveness of DEPTS on handling PTS. In most cases, DEPTS achieves significant improvements over the best baseline. Specifically, the error reduction can even reach up to 20% for a few cases. All codes are publicly available at https://github.com/weifantt/DEPTS.

## 1 Introduction

Time series (TS) with apparent periodic (seasonal) oscillations, referred to as *periodic time series* (PTS) in this paper, is pervasive in a wide range of critical industries, such as seasonal electricity spot prices in power industry (Koopman et al., 2007), periodic traffic flows in transportation (Lippi et al., 2013), periodic carbon dioxide exchanges and water flows in sustainability domain (Seymour, 2001; Tesfaye et al., 2006). Apparently, PTS forecasting plays a crucial role in these industries since it can foster their business development by facilitating a variety of capabilities, including early warning, pre-planning, and resource scheduling (Kahn, 2003; Jain, 2017).

Given the pervasiveness and importance of PTS, two obstacles, however, largely hinder the performance of existing forecasting models. First, future TS signals yield complicated dependencies on both adjacent historical observations and inherent periodicity. Nevertheless, many existing studies did not consider this distinctive periodic property (Salinas et al., 2020; Toubeau et al., 2018; Wang et al., 2019; Oreshkin et al., 2020). The performance of these methods has been greatly restrained due to its ignorance of periodicity modeling. Some other efforts, though explicitly introducing periodicity modeling, only followed some arbitrary yet simple assumptions, such as additive or multiplicative seasonality, to capture certain plain periodic effects (Holt, 1957; 2004; Vecchia, 1985b; Taylor & Letham, 2018). These methods failed to model complicated periodic dependencies beyond much simplified assumptions. The second challenge lies in that the inherent periodicity of a typical real-world TS is usually composed of various periods with different amplitudes and frequencies. For example, Figure 1 exemplifies the sophisticated composition of diversified periods via

---

*Work is done during the internship at Microsoft Research.

Figure 1: We visualize the electricity load TS in a region of California to show diversified periods. In the upper part, we depict the whole TS with the length of eight years, and in the bottom part, we plot three segments with the lengths of half year, one month, and one week, respectively.

a real-world eight-years hourly TS of electricity load in a region of California. However, existing methods (Taylor & Letham, 2018; Smyl, 2020) required the pre-specification of periodic frequencies before estimating other parameters from data, which attempted to evade this obstacle by transferring the burden of periodicity coefficient initialization to practitioners.

To better tackle the aforementioned two challenges, we develop a *deep expansion* learning framework, DEPTS, for *PTS* forecasting. The core idea of DEPTS is to build a deep neural network that conducts the progressive expansions of the complicated dependencies of PTS signals on periodicity to facilitate forecasting. We start from a novel decoupled formulation for PTS forecasting by introducing the periodic state as a hidden variable. This new formulation stimulates us to make more customized and dedicated designs to handle the two specific challenges mentioned above.

For the first challenge, we develop an expansion module on top of residual learning (He et al., 2016; Oreshkin et al., 2020) to conduct layer-by-layer expansions between observed TS signals and hidden periodic states. With such a design, we can build a deep architecture with both high capacities and efficient parameter optimization to model those complicated dependencies of TS signals on periodicity. For the second challenge, we build a periodicity module to estimate the periodic states from observational data. We represent the hidden periodic state with respect to time as a parameterized periodic function with sufficient expressiveness. In this work, for simplicity, we instantiate this function as a series of cosine functions. To release the burden of manually setting periodic coefficients for different data, we develop a data-driven parameter initialization strategy on top of Discrete Cosine Transform (Ahmed et al., 1974). After that, we combine the periodicity module with the expansion module to perform end-to-end learning.

To the best of our knowledge, DEPTS is a very early attempt to build a customized deep learning (DL) architecture for PTS that explicitly takes account of the periodic property. Moreover, with two delicately designed modules, DEPTS also owns certain interpretable capabilities. First, the expansions of forecasts can distinguish the contributions from either adjacent TS signals or inherent periodicity, which intuitively illustrate how the future TS signals may vary based on local momenta and global periodicity. Second, coefficients of the periodicity module have their own practical meanings, such as amplitudes and frequencies, which provide certain interpretable effects inherently.

We conduct experiments on both synthetic data and real-world data, which all demonstrate the superiority of DEPTS on handling PTS. On average, DEPTS reduces the error of the best baseline by about 10%. In a few cases, the error reduction can even reach up to 20%. Besides, we also include extensive ablation tests to verify our critical designs and visualize specific model components to interpret model behaviors.

## 2 RELATED WORK

TS forecasting is a longstanding research topic that has been extensively studied for decades. After a comprehensive review of the literature, we find three types of paradigms in developing TS models. At an early stage, researchers developed simple yet effective statistical modeling approaches, including exponentially weighted moving averages (Holt, 1957; 2004; Winters, 1960), auto-regressive moving averages (ARMA) (Whittle, 1951; 1963), the unified state-space modeling approach as well as other various extensions (Hyndman & Khandakar, 2008). However, these statistical approaches only considered the linear dependencies of future TS signals on past observations. To handle high-order dependencies, researchers attempted to adopt a hybrid design that combines statistical modeling with more advanced high-capacity models (Montero-Manso et al., 2020; Smyl, 2020). At the

same time, with the great successes of DL in computer vision (He et al., 2016) and natural language processing (Vaswani et al., 2017), various DL models have also been developed for TS forecasting (Rangapuram et al., 2018; Toubeau et al., 2018; Salinas et al., 2020; Zia & Razzaq, 2020; Cao et al., 2020). Among them, the most representative one is N-BEATS (Oreshkin et al., 2020), which is a pure DL architecture that has achieved state-of-the-art performance across a wide range of benchmarks. The connections between DEPTS and N-BEATS have been discussed in Section 4.2.

As for PTS forecasting, many traditional statistical approaches explicitly considered the periodic property, such as periodic ARMA (PARMA) (Vecchia, 1985a;b) and its variants (Tesfaye et al., 2006; Anderson et al., 2007; Dudek et al., 2016). However, as discussed in Sections 1 and 3, these methods only followed some arbitrary yet simple assumptions, such as additive or multiplicative seasonality, and thus cannot well handle complicated periodic dependencies in many real-world scenarios. Besides, other recent studies either followed the similar assumptions for periodicity or required the pre-specification of periodic coefficients (Taylor & Letham, 2018; Smyl, 2020). To the best of our knowledge, we are the first work that develops a customized DL architecture to model complicated periodic dependencies and to capture diversified periodic compositions simultaneously.

## 3 PROBLEM FORMULATIONS

We consider the point forecasting problem of regularly sampled uni-variate TS. Let $x_t$ denote the time series value at time-step $t$, and the classical auto-regressive formulation is to project the historical observations $\boldsymbol{x}_{t-L:t} = [x_{t-L}, \ldots, x_{t-1}]$ into its subsequent future values $\boldsymbol{x}_{t:t+H} = [x_t, \ldots, x_{t+H-1}]$:

$$\boldsymbol{x}_{t:t+H} = \mathcal{F}_\Theta(\boldsymbol{x}_{t-L:t}) + \boldsymbol{\epsilon}_{t:t+H}, \tag{1}$$

where $H$ is the length of the forecast horizon, $L$ is the length of the lookback window, $\mathcal{F}_\Theta : \mathbb{R}^L \to \mathbb{R}^H$ is a mapping function parameterized by $\Theta$, and $\boldsymbol{\epsilon}_{t:t+H} = [\epsilon_t, \ldots, \epsilon_{t+H-1}]$ denotes a vector of independent and identically distributed Gaussian noises. Essentially, the fundamental assumption behind this formulation is the Markov property $\boldsymbol{x}_{t:t+H} \perp \boldsymbol{x}_{0:t-L} | \boldsymbol{x}_{t-L:t}$, which assumes that the future values $\boldsymbol{x}_{t:t+H}$ are independent of all farther historical values $\boldsymbol{x}_{0:t-L}$ given the adjacent short-term observations $\boldsymbol{x}_{t-L:t}$. Note that most existing DL models (Salinas et al., 2020; Toubeau et al., 2018; Wang et al., 2019; Oreshkin et al., 2020) directly follow this formulation to solve TS. Even traditional statistical TS models (Holt, 1957; 2004; Winters, 1960) are indeed consistent with that if omitting those long-tail exponentially decayed dependencies introduced by moving averages.

To precisely formulate PTS, on the other hand, this assumption needs to be slightly modified such that the dependency of $\boldsymbol{x}_{t:t+H}$ on $\boldsymbol{x}_{t-L:t}$ is further conditioned on the inherent periodicity, which can be anchored by associated time-steps. Accordingly, we alter the equation (1) into

$$\boldsymbol{x}_{t:t+H} = \mathcal{F}_\Theta^{'}(\boldsymbol{x}_{t-L:t}, t) + \boldsymbol{\epsilon}_{t:t+H}, \tag{2}$$

where other than $\boldsymbol{x}_{t-L:t}$, $\mathcal{F}_\Theta^{'} : \mathbb{R}^L \times \mathbb{R} \to \mathbb{R}^H$ takes an extra argument $t$, which denotes the forecasting time-step. Existing methods for PTS adopt a few different instantiations of $\mathcal{F}_\Theta^{'}$. For example, Holt (1957; 2004) developed several exponentially weighted moving average processes with additive or multiplicative seasonality. Vecchia (1985a;b) adopted the multiplicative seasonality by treating the coefficients of the auto-regressive moving average process as time dependent. Smyl (2020) also adopted the multiplicative seasonality and built a hybrid method by coupling that with recurrent neural networks (Hochreiter & Schmidhuber, 1997), while Taylor & Letham (2018) chose the additive seasonality by adding the periodic forecast with other parts as the final forecast.

## 4 DEPTS

In this section, we elaborate on our new framework, DEPTS. First, we start with a decoupled formulation of (2) in Section 4.1. Then, we illustrate the proposed neural architecture for this formulation in Sections 4.2 and 4.3. Last, we discuss the interpretable capabilities in Section 4.4.

### 4.1 THE DECOUPLED FORMULATION

To explicitly tackle the two-sided challenges of PTS forecasting, i.e., complicated periodic dependencies and diversified periodic compositions, we introduce a decoupled formulation (3) that refines

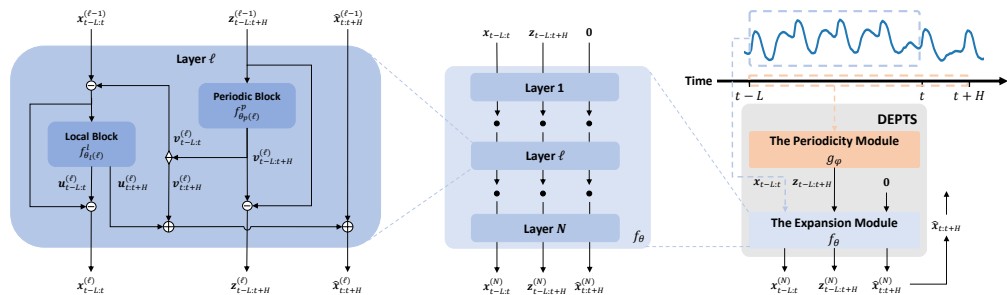

Figure 2: In the right part, we visualize the overall data flows for our framework, DEPTS. In the middle part, we plot the integral structure of three layer-by-layer expansion branches in the expansion module $f_\theta$. In the left part, we depict the detailed residual connections within a single layer.

(2) by introducing a hidden variable $z_t$ to represent the periodic state at time-step $t$:

$$\boldsymbol{x}_{t:t+H} = f_\theta(\boldsymbol{x}_{t-L:t}, \boldsymbol{z}_{t-L:t+H}) + \boldsymbol{\epsilon}_{t:t+H}, \quad z_t = g_\phi(t), \tag{3}$$

where we treat $z_t \in \mathbb{R}^1$ as a scalar value to be consistent with the uni-variate TS $x_t \in R^1$, we use $f_\theta : \mathbb{R}^L \times \mathbb{R}^{L+H} \to \mathbb{R}^H$ to model complicated dependencies of the future signals $\boldsymbol{x}_{t:t+H}$ on the local observations $\boldsymbol{x}_{t-L:t}$ and the corresponding periodic states $\boldsymbol{z}_{t-L:t+H}$ within the lookback and forecast horizons, and $g_\phi : \mathbb{R}^1 \to \mathbb{R}^1$ is to produce a periodic state $z_t$ for a specific time-step $t$. The right part of Figure 2 depicts the overall data flows of this formulation, in which the expansion module $f_\theta$ and the periodicity module $g_\phi$ are responsible for handling the two aforementioned PTS-specific challenges, respectively.

## 4.2 THE EXPANSION MODULE

To effectively model complicated periodic dependencies, the main challenge lies in the trade-off between model capacity and generalization. To avoid the over-fitting issue, many existing PTS approaches relied on the assumptions of additive or multiplicative seasonality (Holt, 1957; Vecchia, 1985b; Anderson et al., 2007; Taylor & Letham, 2018; Smyl, 2020), which however can hardly express periodicity beyond such simplified assumptions. Lately, residual learning has shown its great potentials in building expressive and generalizable DL architectures for a variety of crucial applications, such as computer vision (He et al., 2016) and language understanding (Vaswani et al., 2017). Specifically, N-BEATS (Oreshkin et al., 2020) conducted a pioneer demonstration of introducing residual learning to TS forecasting. Inspired by these successful examples and with full consideration of PTS-specific challenges, we develop a novel expansion module $f_\theta$ on top of residual learning to characterize the complicated dependencies of $\boldsymbol{x}_{t:t+H}$ on $\boldsymbol{x}_{t-L:t}$ and $\boldsymbol{z}_{t-L:t+H}$.

The proposed architecture for $f_\theta$, as shown in the middle part of Figure 2, consists of $N$ layers in total. As further elaboration in the left part of Figure 2, each layer $\ell$, share an identical residual structure consisting of three residual branches, which correspond to the recurrence relations of $\boldsymbol{z}_{t-L:t+H}^{(\ell)}$, $\boldsymbol{x}_{t-L:t}^{(\ell)}$, and $\hat{\boldsymbol{x}}_{t:t+H}^{(\ell)}$, respectively. Here $\boldsymbol{x}_{t-L:t}^{(\ell)}$ and $\boldsymbol{z}_{t-L:t+H}^{(\ell)}$ denote the residual terms of $\boldsymbol{x}_{t-L:t}$ and $\boldsymbol{z}_{t-L:t+H}$ after $\ell$-layers expansions, and $\hat{\boldsymbol{x}}_{t:t+H}^{(\ell)}$ denotes the cumulative forecasts after $\ell$ layers. In layer $\ell$, three residual branches are specified by two parameterized blocks, a local block $f_{\theta_l(\ell)}^l$ and a periodic block $f_{\theta_p(\ell)}^p$, where $\theta_l(\ell)$ and $\theta_p(\ell)$ are their respective parameters.

First, we present the updating equation for $\boldsymbol{z}_{t-L:t+H}^{(\ell-1)}$, which aims to produce the forecasts from periodic states and exclude the periodic effects that have been used. To be more concrete, $f_{\theta_p(\ell)}^p$ takes in $\boldsymbol{z}_{t-L:t+H}^{(\ell-1)}$ and emits the $\ell$-th expansion term of periodic states, denoted as $\boldsymbol{v}_{t-L:t+H}^{(\ell)} \in \mathbb{R}^{L+H}$. $\boldsymbol{v}_{t-L:t+H}^{(\ell)}$ has two components, a backcast component $\boldsymbol{v}_{t-L:t}^{(\ell)}$ and a forecast one $\boldsymbol{v}_{t:t+H}^{(\ell)}$. We leverage $\boldsymbol{v}_{t-L:t}^{(\ell)}$ to exclude the periodic effects from $\boldsymbol{x}_{t-L:t}^{(\ell-1)}$ and adopt $\boldsymbol{v}_{t:t+H}^{(\ell)}$ as the portion of forecasts purely from the $\ell$-th periodic block. Besides, when moving to the next layer, we exclude $\boldsymbol{v}_{t-L:t+H}^{(\ell)}$ from

$z_{t-L:t+H}^{(\ell-1)}$ as $z_{t-L:t+H}^{(\ell)} = z_{t-L:t+H}^{(\ell-1)} - v_{t-L:t+H}^{(\ell)}$ to encourage the subsequent periodic blocks to focus on the unresolved residue $z_{t-L:t+H}^{(\ell)}$.

Then, since $v_{t-L:t}^{(\ell)}$ is related to the periodic components that have been used to produce a part of forecasts, we construct the input to $f_{\theta_l(\ell)}^l$ as $(x_{t-L:t}^{(\ell-1)} - v_{t-L:t}^{(\ell)})$. Here the purpose is to encourage $f_{\theta_l(\ell)}^l$ to focus on the unresolved patterns within $x_{t-L:t}^{(\ell-1)}$. $f_{\theta_l(\ell)}^l$ emits $u_{t-L:t}^{(\ell)}$ and $u_{t:t+H}^{(\ell)}$, which correspond to the local backcast and forecast expansion terms of the $\ell$-th layer, respectively. After that, we update $x_{t-L:t}^{(\ell)}$ by further subtracting $u_{t-L:t}^{(\ell)}$ from $(x_{t-L:t}^{(\ell-1)} - v_{t-L:t}^{(\ell)})$ as $x_{t-L:t}^{(\ell)} = x_{t-L:t}^{(\ell-1)} - v_{t-L:t}^{(\ell)} - u_{t-L:t}^{(\ell)}$. Here the insight is also to exclude all analyzed patterns of this layer to let the following layers focus on unresolved information. Besides, we update $\hat{x}_{t:t+H}^{(\ell)}$ by adding both $u_{t:t+H}^{(\ell)}$ and $v_{t:t+H}^{(\ell)}$ as $\hat{x}_{t:t+H}^{(\ell)} = \hat{x}_{t:t+H}^{(\ell-1)} + u_{t:t+H}^{(\ell)} + v_{t:t+H}^{(\ell)}$. The motivation of such expansion is to decompose the forecasts from the $\ell$-th layer into two parts, $u_{t:t+H}^{(\ell)}$ and $v_{t:t+H}^{(\ell)}$, which correspond to the part from local observations excluding redundant periodic information and the other part purely from periodic states, respectively.

Note that before the first layer, we have $x_{t-L:t}^{(0)} = x_{t-L:t}$, $z_{t-L:t+H}^{(0)} = z_{t-L:t+H}$, and $\hat{x}_{t:t+H}^{(0)} = \mathbf{0}$. Besides, we collect the cumulative forecasts $\hat{x}_{t:t+H}^{(N)}$ of the $N$-th layer as the overall forecasts $\hat{x}_{t:t+H}$. Therefore, after stacking $N$ layers of $z_{t-L:t+H}^{(\ell)}$, $x_{t-L:t}^{(\ell)}$, and $\hat{x}_{t:t+H}^{(\ell)}$, we have the following triply residual expansions that encapsulate the left and middle parts of Figure 2:

$$z_{t-L:t+H} = z_{t-L:t+H}^{(0)} = \sum_{\ell=1}^{N} v_{t-L:t+H}^{(\ell)} + z_{t-L:t+H}^{(N)},$$

$$x_{t-L:t} = x_{t-L:t}^{(0)} = \sum_{\ell=1}^{N} (u_{t-L:t}^{(\ell)} + v_{t-L:t}^{(\ell)}) + x_{t-L:t}^{(N)}, \qquad (4)$$

$$\hat{x}_{t:t+H} = \hat{x}_{t:t+H}^{(N)} = \sum_{\ell=1}^{N} (u_{t:t+H}^{(\ell)} + v_{t:t+H}^{(\ell)}),$$

where $z_{t-L:t+H}^{(N)}$ and $x_{t-L:t}^{(N)}$ are deemed to be the residues irrelevant to forecasting.

**Connections and differences to N-BEATS.** Our design of $f_\theta$ shares the similar insight with N-BEATS (Oreshkin et al., 2020), which is stimulating a deep neural network to learn expansions of raw TS signals progressively, whereas N-BEATS only considered the generic TS by modeling the dependencies of $x_{t:t+H}$ on $x_{t-L:t}$. In contrast, our design is to capture the complicated dependencies of $x_{t:t+H}$ on $x_{t-L:t}$ and $z_{t-L:t+H}$ for PTS. Moreover, to achieve periodicity modeling, N-BEATS produces coefficients solely based on the input signals within a lookback window for a group of predefined seasonal basis vectors with fixed frequencies and phases. However, our work can capture diversified periods in practice and model the inherent global periodicity.

**Inner architectures of local and periodic blocks.** The local block $f_{\theta_l(\ell)}^l$ aims to produce a part of forecasts based on the local observations excluding redundant periodic information as $(x_{t-L:t} - \sum_{i=1}^{\ell-1} v_{t-L:t}^{(i)})$. Thus, we reuse the generic block developed by Oreshkin et al. (2020), which consists of a series of fully connected layers. As for the periodic block $f_{\theta_p(\ell)}^p$, which handles the relatively stable periodic states, we can adopt a simple two-layer perception. Due to the space limit, we include more details of inner block architectures in Appendix A.

## 4.3 THE PERIODICITY MODULE

To represent the sophisticated periodicity composed of various periodic patterns, we estimate $z_t$ via a parameterized periodic function $g_\phi(t)$ that holds sufficient capacities to incorporate diversified periods. In this work, for simplicity, we instantiate this function as a series of cosine functions as $g_\phi(t) = A_0 + \sum_{k=1}^{K} A_k \cos(2\pi F_k t + P_k)$, where $K$ is a hyper-parameter denoting the total number

of periods, $A_0$ is a scalar parameter for the base scale, $A_k$, $F_k$, and $P_k$ are the scalar parameters for the amplitude, the frequency, and the phase of the $k$-th cosine function, respectively, and $\phi$ represents the set of all parameters. Coupling $g_\phi$ with $f_\theta$ illustrated in Section 4, we can effectively model the periodicity-aware auto-regressive forecasting process in the equation (2). However, it is extremely challenging to directly conduct the joint optimization of $\phi$ and $\theta$ from random initialization. The reason is that in such a highly non-convex condition, the coefficients in $\phi$ are easily trapped into numerous local optima, which do not necessarily characterize our desired periodicity.

**Parameter Initialization.** To overcome the optimization obstacle mentioned above, we formalize a two-stage optimization problem based on raw PTS signals to find good initialization for $\phi$. First, we construct a surrogate function, $g_\phi^M(t) = A_0 + \sum_{k=1}^{K} M_k \cdot A_k cos(2\pi F_k t + P_k)$, to enable the selection of a subset of periods via $M = \{M_k, k \in \{1, \cdots, K\}\}$, where each $M_k \in \{0, 1\}$ is a mask variable to enable or disable certain periods. Note that $g_\phi(t)$ is equivalent to $g_\phi^M(t)$ when every $M_k$ is equal to one. Then, we construct the following two-stage optimization problem:

$$M^* = \underset{\|M\|_1 <= J}{\arg\min} \, \mathcal{L}_{D_{val}}(g_{\phi^*}^M(t)), \quad \phi^* = \underset{\phi}{\arg\min} \, \mathcal{L}_{D_{train}}(g_\phi(t)), \tag{5}$$

where $\mathcal{L}_{D_{train}}$ and $\mathcal{L}_{D_{val}}$ denote the discrepancy losses on training and validation, respectively; the inner stage is to obtain $\phi^*$ that minimizes the discrepancy between $z_t$ and $x_t$ on the training data $D_{train}$; the outer stage is a binary integer programming on the validation data $D_{val}$ to find $M^*$ that can select certain periods with good generalization, and the hyper-parameter $J$ controls the maximal number of periods being selected. With the help of such two-stage optimization, we are able to estimate generalizable periodic coefficients from observational data as a good starting point for $\phi$ to be jointly optimized with $\theta$. Nevertheless, it is still costly to perform exact optimization of equations (5) in practice. Thus, we develop a fast approximation algorithm to obtain an acceptable solution with affordable costs. Our approximation algorithm contains the following two steps: 1) conducting Discrete Cosine Transform (Ahmed et al., 1974) of PTS signals on $D_{train}$ to select top-$K$ cosine bases with the largest amplitude as an approximated solution of $\phi^*$; 2) iterating over the selected $K$ cosine bases from the largest amplitude to the smallest one and greedily select $J$ periods that generalize well on the validation set. Due to the space limit, we include more details of this approximation algorithm in Appendix B. After obtaining approximated solutions $\tilde{\phi}^*$ and $\tilde{M}^*$, we fix $M = \tilde{M}^*$ to exclude those unstable periodic coefficients and initialize $\phi$ with $\tilde{\phi}^*$ to avoid being trapped into bad local optima. Then, we follow the formulation (3) to perform the joint learning of $\phi$ and $\theta$ in an end-to-end manner.

## 4.4 Interpretability

Owing to the specific designs of $f_\theta$ and $g_\phi$, our architecture is born with a degree of interpretability. First, for $f_\theta$, as shown in equations (4), we decompose $\hat{x}_{t:t+H}$ into two types of components, $u_{t:t+H}^{(\ell)}$ and $v_{t:t+H}^{(\ell)}$. Note that $v_{t:t+H}^{(\ell)}$ is conditioned on $z_{t-L:t+H}$ and independent of $x_{t-L:t}$. Thus, $\sum_{\ell=1}^{N} v_{t:t+H}^{(\ell)}$ represents the portion of forecasts purely from periodic states. Meanwhile, $u_{t:t+H}^{(\ell)}$ depends on both $x_{t-L:t}$ and $z_{t-L:t+H}$, and it is transformed by feeding the subtraction of $v_{t-L:t}^{(\ell)}$ from $x_{t-L:t}^{(\ell-1)}$ into the $\ell$-th local block. Thus, we can regard $\sum_{\ell=1}^{N} u_{t-L:t}^{(\ell)}$ as the forecasts from the local historical observations excluding the periodic effects, referred to as the local momenta in this paper. In this way, we can differentiate the contribution to the final forecasts into both the global periodicity and the local momenta. Second, $g_\phi$, the periodicity estimation module in our architecture, also has interpretable effects. Specifically, the coefficients in $g_\phi(t)$ have practical meanings, such as amplitudes, frequencies, and phases. We can interpret these coefficients as the inherent properties of the series and connect them to practical scenarios. Furthermore, by grouping various periods together, $g_\phi$ provides us with the essential periodicity of TS filtering out various local momenta.

## 5 Experiments

Our empirical studies aim to answer three questions. 1) Why is it important to model the complicated dependencies of PTS signals on its inherent periodicity? 2) How much benefit can DEPTS gain for PTS forecasting compared with existing state-of-the-art models? 3) What kind of interpretability can

Figure 3: Performance comparisons of N-BEATS and DEPTS (ours) on synthetic data, in which we simulate different periodic dependencies, such as linear, quadratic, and cubic.

DEPTS offer based on our two customized modules, $f_\theta$ and $g_\phi$? To answer the first two questions, we conduct extensive experiments on both synthetic data and real-world data, which are illustrated in Section 5.1 and 5.2, respectively. Then, Section 4.4 answers the third question by comparing and interpreting model behaviors for specific cases.

**Baselines.** We adopt the state-of-the-art DL architecture, N-BEATS (Oreshkin et al., 2020), as our primary baseline since it has been shown to outperform a wide range of DL models, including MatFact (Yu et al., 2016), Deep State (Rangapuram et al., 2018), Deep Factors (Wang et al., 2019), and DeepAR (Salinas et al., 2020), and many competitive hybrid methods (Montero-Manso et al., 2020; Smyl, 2020). Besides, we also include PARMA as a reference to be aware of the positions of conventional statistical models. For this baseline, we leverage the AutoARIMA implementation provided by Löning et al. (2019) to search for the best configurations automatically.

**Evaluation Metrics.** To compare different models, we utilize the following two metrics, normalized deviation, abbreviated as *nd*, and normalized root-mean-square error, denoted as *nrmse*, which are conventionally adopted by Yu et al. (2016); Rangapuram et al. (2018); Salinas et al. (2020); Oreshkin et al. (2020) on PTS-related benchmarks.

$$nd = \frac{\frac{1}{|\Omega|}\sum_{(i,t)\in\Omega}|x_t^i - \hat{x}_t^i|}{\frac{1}{|\Omega|}\sum_{(i,t)\in\Omega}|x_t^i|}, \quad nrmse = \frac{\sqrt{\frac{1}{|\Omega|}\sum_{(i,t)\in\Omega}(x_t^i - \hat{x}_t^i)^2}}{\frac{1}{|\Omega|}\sum_{(i,t)\in\Omega}|x_t^i|}, \quad (6)$$

where $i$ is the index of TS in a dataset, $t$ is the time index, and $\Omega$ denotes the whole evaluation space.

## 5.1 EVALUATION ON SYNTHETIC DATA

To intuitively illustrate the importance of periodicity modeling, we generate synthetic data with various periodic dependencies and multiple types of periods. Specifically, we generate a simulated TS signal $x_t$ by composing an auto-regressive signal $l_t$, corresponding to the local momentum, and a compounded periodic signal $p_t$, denoting the global periodicity, via a function $f^c$ as $x_t = f^c(l_t, p_t)$, which characterizes the dependency of $x_t$ on $l_t$ and $p_t$. First, we produce $l_t$ via an auto-regressive process, $l_t = \sum_{i=1}^{L}\alpha_i l_{t-i} + \epsilon_t^l$, in which $\alpha_i$ is a coefficient for the $i$-lag dependency, and the error term $\epsilon_t^l \sim \mathcal{N}(0, \sigma^l)$ follows a zero-mean Gaussian distribution with standard deviation $\sigma^l$. Then, we produce $p_t$ by sampling from another Gaussian distribution $\mathcal{N}(z_t, \sigma^p)$, in which $z_t$ is characterized by a periodic function (instantiated as $g_\phi(t)$ in Section 4.3), and $\sigma^p$ is a standard deviation to adjust the degree of dispersion for periodic samples. Next, we take three types of $f^c(l_t, p_t)$, $(l_t + p_t)$, $(l_t + p_t)^2$, and $(l_t + p_t)^3$, to characterize the linear, quadratic, and cubic dependencies of $x_t$ on $l_t$ and $p_t$, respectively. Last, after data generation, all models only have access to the final mixed signal $x_t$ for training and evaluation.

Due to the space limit, we include the main results in Figure 3 and leave finer grained parameter specifications and more experimental details to Appendix C. For each setup (linear, quadratic, cubic) in Figure 3, we have searched for the best lookback length ($L$) for N-BEATS and the best number of periods ($J$) for DEPTS on the validation set and re-run the model training with five different random seeds to produce robust results on the test set. We can observe that for all cases, even with an exhaustive search of proper lookback lengths for N-BEATS, there exists a considerable performance gap between it and DEPTS, which verifies the utility of explicit periodicity modeling. Moreover, as the periodic dependency becomes more complex (from linear to cubic), the average error reduction of DEPTS over N-BEATS keeps increasing (from 7% to 11%), which further demonstrates the importance of modeling high-order periodic effects.

Table 1: Performance comparisons (*nd*) on ELECTRICITY, TRAFFIC, and M4 (HOURLY). For the first two, we follow two different test splits defined in previous studies.

| Model | ELECTRICITY | | TRAFFIC | | M4 (HOURLY) |
|---|---|---|---|---|---|
| | 2014-09-01 | 2014-12-25 | 2008-06-15 | 2009-03-24 | |
| MatFact | 0.16 | 0.255 | 0.20 | 0.187 | n/a |
| DeepAR | 0.07 | n/a | 0.17 | n/a | 0.09 |
| Deep State | 0.083 | n/a | 0.167 | n/a | 0.044 |
| N-BEATS | 0.064 | 0.171 | 0.114 | 0.112 | 0.023 |
| DEPTS | **0.060** | **0.139** | **0.111** | **0.107** | **0.021** |

Table 2: Performance comparisons (*nd* and *nrmse*) on CAISO and NP, where we define four test splits to cover all four seasons of the last year for each benchmark.

| Dataset | Model | 2020-01-01 | | 2020-04-01 | | 2020-07-01 | | 2020-10-01 | |
|---|---|---|---|---|---|---|---|---|---|
| | | *nd* | *nrmse* | *nd* | *nrmse* | *nd* | *nrmse* | *nd* | *nrmse* |
| CAISO | PARMA | 0.089 | 0.169 | 0.107 | 0.214 | 0.116 | 0.215 | 0.079 | 0.148 |
| | N-BEATS | 0.029 | 0.058 | 0.031 | 0.073 | 0.030 | 0.064 | 0.026 | 0.057 |
| | DEPTS | **0.024** | **0.049** | **0.028** | **0.063** | **0.029** | **0.058** | **0.020** | **0.042** |
| NP | PARMA | 0.220 | **0.350** | 0.201 | 0.321 | 0.216 | 0.352 | 0.199 | 0.305 |
| | N-BEATS | 0.207 | 0.434 | 0.154 | 0.237 | 0.195 | 0.315 | 0.211 | 0.332 |
| | DEPTS | **0.196** | 0.377 | **0.145** | **0.224** | **0.169** | **0.269** | **0.179** | **0.281** |

## 5.2 EVALUATION ON REAL-WORLD DATA

Other than simulation experiments, we further demonstrate the effectiveness of DEPTS on real-world data. We adopt three existing PTS-related datasets, ELECTRICITY[1], TRAFFIC[2], and M4 (HOURLY)[3], which contain various long-term (quarterly, yearly), mid-term (monthly, weekly), and short-term (daily, hourly) periodic effects corresponding to regular economic and social activities. These datasets serve as common benchmarks for many recent studies (Yu et al., 2016; Rangapuram et al., 2018; Salinas et al., 2020; Oreshkin et al., 2020). For ELECTRICITY and TRAFFIC, we follow two different test splits defined by Salinas et al. (2020) and Yu et al. (2016), and the evaluation horizon covers the first week starting from the split date. As for M4 (HOURLY), we adopt the official test set. Besides, we note that the time horizons covered by these three benchmarks are still too short, which results in very limited data being left for periodicity learning if we alter the time split too early. This drawback of lacking enough long PTS limits the power of periodicity modeling and thus may hinder the research development in this field. To further verify the importance of periodicity modeling in real-world scenarios, we construct two new benchmarks with sufficiently long PTS from public data sources. The first one, denoted as CAISO, contains eight-years hourly actual electricity load series in different zones of California[4]. The second one, referred to as NP, includes eight-years hourly energy production volume series in multiple European countries[5]. Accordingly, we define four test splits that correspond to all four seasons of the last year for robust evaluation.

For all benchmarks, we search for the best hyper-parameters of DEPTS on the validation set. Similar to N-BEATS (Oreshkin et al., 2020), we also produce ensemble forecasts of multiple models trained with different lookback lengths and random initialization seeds. Tables 1 and 2 show the overall performance comparisons. On average, the error reductions (*nd*) of DEPTS over N-BEATS on ELECTRICITY, TRAFFIC, M4 (HOURLY), CAISO, and NP are 12.5%, 3.5%, 8.7%, 13.3%, and 9.9%, respectively. Interestingly, we observe some prominent improvements in a few specific cases,

---

[1]https://archive.ics.uci.edu/ml/datasets/ElectricityLoadDiagrams20112014
[2]https://archive.ics.uci.edu/ml/datasets/PEMS-SF
[3]https://github.com/Mcompetitions/M4-methods/tree/master/Dataset/Train
[4]http://www.energyonline.com/Data
[5]https://www.nordpoolgroup.com/Market-data1/Power-system-data

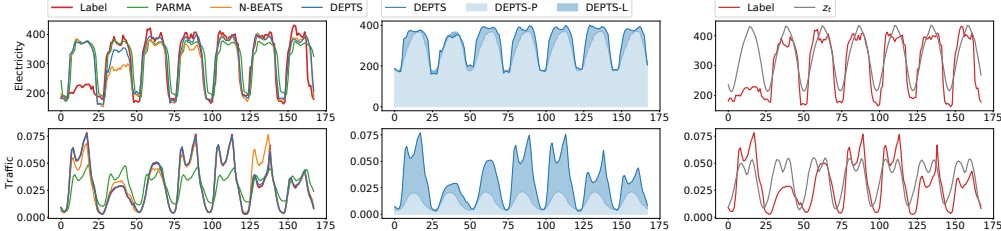

Figure 4: We compare the forecasts of different models (in the left side) and visualize the intermediate states within DEPTS (in the middle and right parts), where DEPTS-P denotes the forecasts from the global periodicity, DEPTS-L denotes the forecasts from the local momenta, and DEPTS is the summation of these two parts, as illustrated in Section 4.2.

such as 18.7% in ELECTRICITY (2014-09-01), 23.1% in CAISO (2020-10-01), and 15.2% in NP (2020-10-01). At the same time, we also observe some tiny improvements, such as 2.6% in TRAFFIC (2008-06-15) and 3.3% in CAISO (2020-07-01). These observations imply that the predictive abilities and the complexities of periodic effects may vary over time, which corresponds to the changes in performance gaps between DEPTS and N-BEATS. Nevertheless, most of the time, DEPTS still brings stable and significant performance gains for PTS forecasting, which clearly demonstrate the importance of periodicity modeling in practice.

Due to the space limit, we leave more details about datasets and hyper-parameters used in real-world experiments to Appendix D. Moreover, to achieve effective periodicity modeling, we have made several critical designs, such as the triply residual expansions in Section 4.2 and the composition of diversified periods in Section 4.3. We also conduct extensive ablation tests to verify these critical designs, which are included in Appendix E.

## 5.3 INTERPRETABILITY

In Figure 4, we illustrate the interpretable effects of DEPTS via two cases, the upper one from ELECTRICITY and the bottom one from TRAFFIC. First, from subplots in the left part, we observe that DEPTS obtains much more accurate forecasts than N-BEATS and PARMA. Then, in the middle and right parts, we can visualize the inner states of DEPTS to interpret how it makes such forecasts. As Section 4.4 states, DEPTS can differentiate the contributions to the final forecasts $\hat{\boldsymbol{x}}_{t:t+H}$ into the local momenta $\sum_{\ell=1}^{N} \boldsymbol{u}_{t:t+H}^{(\ell)}$ and the global periodicity $\sum_{\ell=1}^{N} \boldsymbol{v}_{t:t+H}^{(\ell)}$. Interestingly, we can see that DEPTS has learned two different decomposition strategies: 1) for the upper case, most of the contributions to the final forecasts come from the global periodicity part, which implies that this case follows strong periodic patterns; 2) for the bottom case, the periodicity part just characterizes a major oscillation frequency, while the model relies more on the local momenta to refine the final forecasts. Besides, the right part of Figure 4 depicts the hidden periodic state $z_t$ estimated by our periodicity module $g_\phi(t)$. We can see that $g_\phi(t)$ indeed captures some inherent periodicity. Moreover, the actual PTS signals also present diverse variations at different time, which further demonstrate the importance of leveraging $f_\theta$ to model the dependencies of $\boldsymbol{x}_{t:t+H}$ on both $\boldsymbol{x}_{t-L:t}$ and $\boldsymbol{z}_{t-L:t+H}$. We include more case studies and interpretability analysis in Appendix F.

## 6 CONCLUSION

In this paper, we develop a novel DL framework, DEPTS, for PTS forecasting. Our core contributions are to model complicated periodic dependencies and to capture sophisticated compositions of diversified periods simultaneously. Extensive experiments on both synthetic data and real-world data demonstrate the effectiveness of DEPTS on handling PTS. Moreover, periodicity modeling is actually an old and crucial topic for traditional TS modeling but is rarely studied in the context of DL. Thus we hope that the new DL framework together the two new benchmarks with evident periodicity and sufficiently long observations can facilitate more future research on PTS.

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

# A    BLOCK ARCHITECTURES

As illustrated in Section 4.2, the local block $f^l_{\theta_l(\ell)}$ produces forecasts based on local observed PTS signals excluding redundant periodic effects. This goal aligns with that of N-BEATS to extract informative representations from generic TS signals. Therefore, we reuse the generic block design of N-BEATS to instantiate $f^l_{\theta_l(\ell)}$. Here we include a brief description of the local block for completeness. Please refer to Section 3.1 in (Oreshkin et al., 2020) for more details.

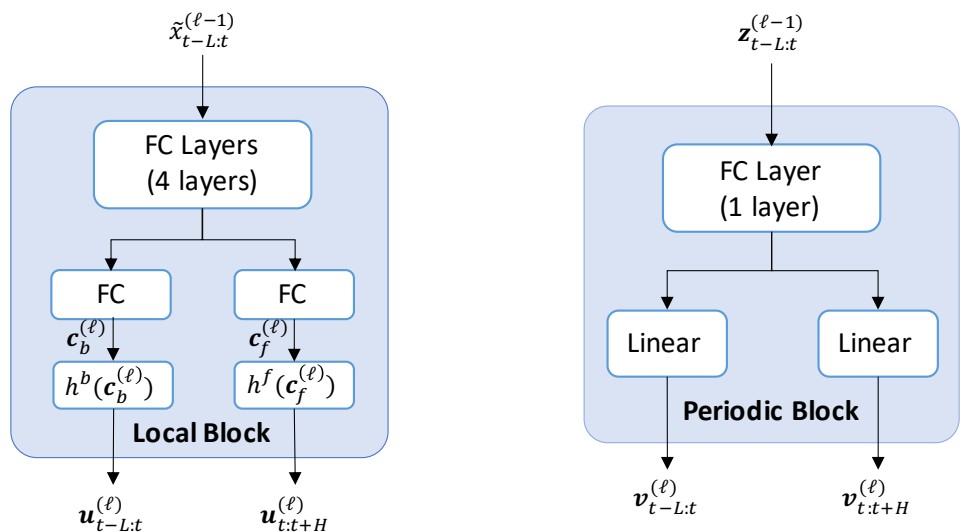

Figure 5: Detailed architectures of the local block and the periodic block in DEPTS.

**Local Block.**    The left part of Figure 5 shows the detailed architecture within a local block, where we use $\tilde{x}^{(\ell)}_{t-L:t} = x^{(\ell-1)}_{t-L:t} - v^{(\ell)}_{t-L:t}$ to denote the portion of the local observations $x^{(\ell-1)}_{t-L:t}$ excluding the periodic effects $v^{(\ell)}_{t-L:t}$ for the $\ell$-th layer. After taking in $\tilde{x}^{(\ell)}_{t-L:t}$, we pass it through four fully-connected layers and then obtain the backcast coefficients $c^{(\ell)}_b$ and the forecast coefficients $c^{(\ell)}_f$ via two linear projections:

$$u^{(\ell),1}_{t-L:t} = \text{FC}_{\ell,1}(\tilde{x}^{(\ell)}_{t-L:t}), \quad u^{(\ell),2}_{t-L:t} = \text{FC}_{\ell,2}(u^{(\ell),1}_{t-L:t}), \quad u^{(\ell),3}_{t-L:t} = \text{FC}_{\ell,3}(u^{(\ell),2}_{t-L:t}),$$
$$u^{(\ell),4}_{t-L:t} = \text{FC}_{\ell,4}(u^{(\ell),3}_{t-L:t}), \quad c^{(\ell)}_b = \text{LINEAR}^b_\ell(u^{(\ell),4}_{t-L:t}), \quad c^{(\ell)}_f = \text{LINEAR}^f_\ell(u^{(\ell),4}_{t-L:t}),$$

where FC is a standard fully-connected layer with ReLU activation (Nair & Hinton, 2010), and LINEAR denotes a linear projection function. Then, we pass these coefficients to the basis layers, $h^b(\cdot)$ and $h^f(\cdot)$, to obtain the backcast term $u^{(\ell)}_{t-L:t}$ and the forecast term $u^{(\ell)}_{t:t+H}$, respectively. The generic choice for $h^b(\cdot)$ can simply be another linear projection function, which is also adopted by us since it produces more competitive and stable performance on PTS-related benchmarks than other interpretable basis layers, as shown by (Oreshkin et al., 2020) in Appendix C.4.

**Periodic Block.**    The periodic block $f^p_{\theta_p(\ell)}$ aims to extract predictive information from associated periodic states, which are relatively simple and stable compared with rapidly shifting PTS signals. Therefore, we can adopt a simple architecture while still maintain desired effects. In this work, we use one-layer standard fully-connected layer to encode $z^{(\ell-1)}_{t-L:t}$ and leverage another two linear projection functions to obtain the backcast term $v^{(\ell)}_{t-L:t}$ and the forecast term $v^{(\ell)}_{t:t+H}$ as the periodic effects of the $\ell$-th layer.

$$v^{(\ell),1}_{t-L:t+H} = \text{FC}_\ell(z^{(\ell-1)}_{t-L:t+H}), \quad v^{(\ell)}_{t-L:t} = \text{LINEAR}^\ell(v^{(\ell),1}_{t-L:t}), \quad v^{(\ell)}_{t:t+H} = \text{LINEAR}^\ell(v^{(\ell),1}_{t:t+H}),$$

where FC and LINEAR share the same meanings mentioned above. Moreover, when training for multiple series simultaneously, we use a series-specific scalar parameter $\alpha_i$ ($i$ is the series index) to take account of differences in the strengths of periodicity by updating $\boldsymbol{v}_{t-L:t+H}^{(\ell)}$ as $\alpha_i \cdot \boldsymbol{v}_{t-L:t+H}^{(\ell)}$.

# B  PARAMETER INITIALIZATION FOR THE PERIODICITY MODULE

As illustrated in Section 4.3, we leverage a fast approximation approach to obtain an acceptable solution of the two-stage optimization problem (5) with affordable costs in practice. Algorithm 1 summarizes the overall procedure for this fast approximation.

---

**Algorithm 1:** Parameter initialization for the periodicity module.

---

**Input:** $D_{train} = \boldsymbol{x}_{0:T_v}$, $D_{val} = \boldsymbol{x}_{T_v:T}$, $K$, and $J$

Conduct DCT over $\boldsymbol{x}_{0:T_v}$.

Sort the top-$K$ cosine bases by amplitudes in descending order to obtain

$\quad \tilde{\phi}^* = \{\tilde{A}_0^*\} \cup \{\tilde{A}_k^*, \tilde{F}_k^*, \tilde{P}_k^*\}_{k=1}^K$.

Initialize $\tilde{M}^* = \boldsymbol{0}$.

**for** $j$ **in** $[1, \cdots, K]$ **do**

$\quad$ **if** $\|\tilde{M}^*\|_1 < J$ **then**

$\quad\quad$ Update $\tilde{M}_j^*$ by $\arg\min_{M_j \in \{0,1\}} \mathcal{L}_{D_{val}}(g_{\tilde{\phi}^*}^{M_j}(t))$

$\quad$ **else**

$\quad\quad$ **return** $\tilde{\phi}^*$ *and* $\tilde{M}^*$

$\quad$ **end**

**end**

**Output:** $\tilde{\phi}^*$ and $\tilde{M}^*$

---

First, we divide the whole PTS signals $\boldsymbol{x}_{0:T}$ into the training part $D_{train} = \boldsymbol{x}_{0:T_v}$ and the validation part $D_{val} = \boldsymbol{x}_{T_v:T}$, where $T_v$ is the split time-step. Then, the inner optimization stage is to identify the optimal parameter set $\phi^*$ that can best fit the training data:

$$\phi^* = \arg\min_{\phi} \mathcal{L}_{D_{train}}(g_\phi(t)), \quad g_\phi(t) = A_0 + \sum_{k=1}^K A_k \cos(2\pi F_k t + P_k), \qquad (7)$$

where the hyper-parameter $K$ controls the capacity of $g_\phi(t)$ and the discrepancy training loss $\mathcal{L}_{D_{train}}$ can be instantiated as the mean square error $\sum_{t=0}^{T_v-1} \|g_\phi(t) - x_t\|_2^2$. Directly optimizing (7) via gradient descent from random initialization is inefficient and time-consuming since it involves numerous gradient updates and is easily trapped into bad local optima. Fortunately, our instantiation of $g_\phi(t)$ as a group of cosine functions shares the similar format with Discrete Cosine Transform (DCT) (Ahmed et al., 1974). Accordingly, we conduct DCT over $\boldsymbol{x}_{0:T_v}$ and select top-$K$ cosine bases with the largest amplitudes, which characterize the major periodic oscillations of this series, as the approximated solution $\tilde{\phi}^*$ of (7).

Next, we enter the outer optimization stage to select certain periods with good generalization:

$$M^* = \arg\min_{\|M\|_1 <= J} \mathcal{L}_{D_{val}}(g_{\phi^*}^M(t)), \quad g_{\phi^*}^M(t) = A_0^* + \sum_{k=1}^K M_k \cdot A_k^* \cos(2\pi F_k^* t + P_k^*), \qquad (8)$$

where the hyper-parameter $J$ further constrains the expressiveness of $g_\phi^M(t)$ for good generalization. Conducting exact optimization of this binary integer programming is also costly since it involves an exponentially growing parameter space. Similarly, to capture the major periodic oscillations as much as possible, we develop a greedy strategy that iterates the selected $K$ cosine bases from the largest amplitude to the smallest and greedily assigns 1 or 0 to $M_k$ depending on whether the $k$-th period further reduces the discrepancy loss on the validation data. Specifically, assuming $K$ periods are already sorted by their amplitudes descendingly and are indexed by $k$, we construct another

surrogate function $g_{\phi^*}^{M_j}(t)$ for the $j$-th greedy step:

$$g_{\phi^*}^{M_j}(t) = M_j \cdot A_j^* \cos(2\pi F_j^* t + P_j^*) + \left[ A_0^* + \sum_{k=1}^{j-1} \tilde{M}_k^* \cdot A_k^* \cos(2\pi F_k^* t + P_k^*) \right], \qquad (9)$$

where $\{\tilde{M}_k^*\}_{k=1}^{j-1}$ is determined in previous steps, $M_j$ is an integer parameter to be set in the current step. Thus, for the $j$-th step, we are actually updating $\tilde{M}_j^*$ by

$$\tilde{M}_j^* = \underset{M_j \in \{0,1\}}{\arg\min} \, \mathcal{L}_{D_{val}}(g_{\phi^*}^{M_j}(t)). \qquad (10)$$

Besides, to tolerate the approximation errors introduced by $\tilde{\phi}^*$, which may result in shifted periodic oscillations, we use Dynamic Time Warping to measure the discrepancy of $g_{\phi^*}^{M_j}(t)$ and $x_t$ on $D_{val}$. We continue this greedily updating steps until selecting $J$ periods in total or completing the traverses of all $K$ selected periods. Finally, we obtain an approximated solution $\tilde{M}^*$ of (8).

**Complexity Analyses.** We also provide the complexity analyses of Algorithm 1, which runs very fast in practice and takes up negligible time compared with training neural networks. Let us denote the length of training series as $L_t$ and the length of validation series as $L_v$. First, the complexity of conducting DCT over training series is $O(L_t log(L_t))$. Then, the complexity of selecting top-K frequencies with the largest amplitudes is $O(L_t log(K))$, which can be ignored since $K << L_t$. Next, we need to select at most $J$ frequencies greedily based on the generalization errors on the validation set. Since we measure the generalization errors via dynamic time warping, the total worst complexity for this selection procedure is $O(KL_v^2)$. In total, the worse complexity of our approximation algorithm for a series is $O(L_t log(L_t) + KL_v^2)$. In practice, $L_v$, the length of the validation series, is relatively small, and $K$, the maximum number of frequencies, can be regarded as a constant. So, the squared complexity term $O(KL_v^2)$ is not a big trouble.

## C  MORE DETAILS ON SYNTHETIC EXPERIMENTS

As Section 5.1 states, we produce a TS $l_t$ via an auto-regressive process, $l_t = \sum_{i=1}^L \alpha_i l_{t-i} + \epsilon_t^l$, in which $\alpha_i$ is a coefficient for the $i$-lag dependence, and the error term $\epsilon_t^l \sim \mathcal{N}(0, \sigma^l)$ follows a zero-mean Gaussian distribution with standard deviation $\sigma^l$. Specifically, we set $L$ as 3 and $\sigma^l$ as 1. We leverage uniform samples from $[-1, 1]$ to initialize $\{\alpha_i\}_{i=1}^3$ and also uniformly sample three values from $[0, 5)$ for the initial points, $l_{-3}$, $l_{-2}$, and $l_{-1}$. Then, we produce $p_t$ by sampling from another Gaussian distribution $\mathcal{N}(z_t, \sigma^p)$, in which $z_t$ is characterized by a periodic function (instantiated as $g_\phi(t)$ in Section 4.3), and $\sigma^p$ is a standard deviation to adjust the degree of dispersion for periodic samples. Specifically, we also set $\sigma^p$ as 1 and produce $z_t$ via a composition of three cosine bases, $8cos(2\pi(t+2)/50)$, $4cos(2\pi(t+3)/10)$, $2cos(2\pi t/4)$, and a base level, 30. These three cosine bases represent long-term, mid-term, short-term periodic effects, respectively, which are very similar to the circumstance in practice. Next, we take three types of $f^c(l_t, p_t)$, $(l_t + p_t)$, $(l_t + p_t)^2$, and $(l_t + p_t)^3$, to characterize the linear, quadratic, and cubic dependencies of $x_t$ on $l_t$ and $p_t$, respectively. We repeat the above procedure for 5000 time steps and divide them into 4000, 100, and 900 for training, validation, and evaluation, respectively. Figure 6 shows the first 1000 time steps of these synthetic series. Note that after data generation, all models only have access to the final mixed signal $x_t$ for training and evaluation.

Moreover, as illustrated in Section 5.1, we search for the best loobkack length ($L$) for N-BEATS and the best number of periods ($J$) for DEPTS. The lookback length for DEPTS is fixed as 3, which is also determined by hyper-parameter tuning on the validation set. Figure 7 shows detailed comparisons of N-BEATS and DEPTS for different configurations of $L$ and $J$. We can see that N-BEATS always needs a relatively long lookback window, such as 48 or 96 time steps, to capture those periodic patterns effectively. Besides, further increasing the lookback length will introduce more irrelevant noises, which overwhelm effective predictive signals and thus result in more worse performance. In contrast, with effective periodicity modeling, DEPTS can achieve better performance by using a short lookback window, which is also consistent with the auto-regressive process that governs the local momenta.

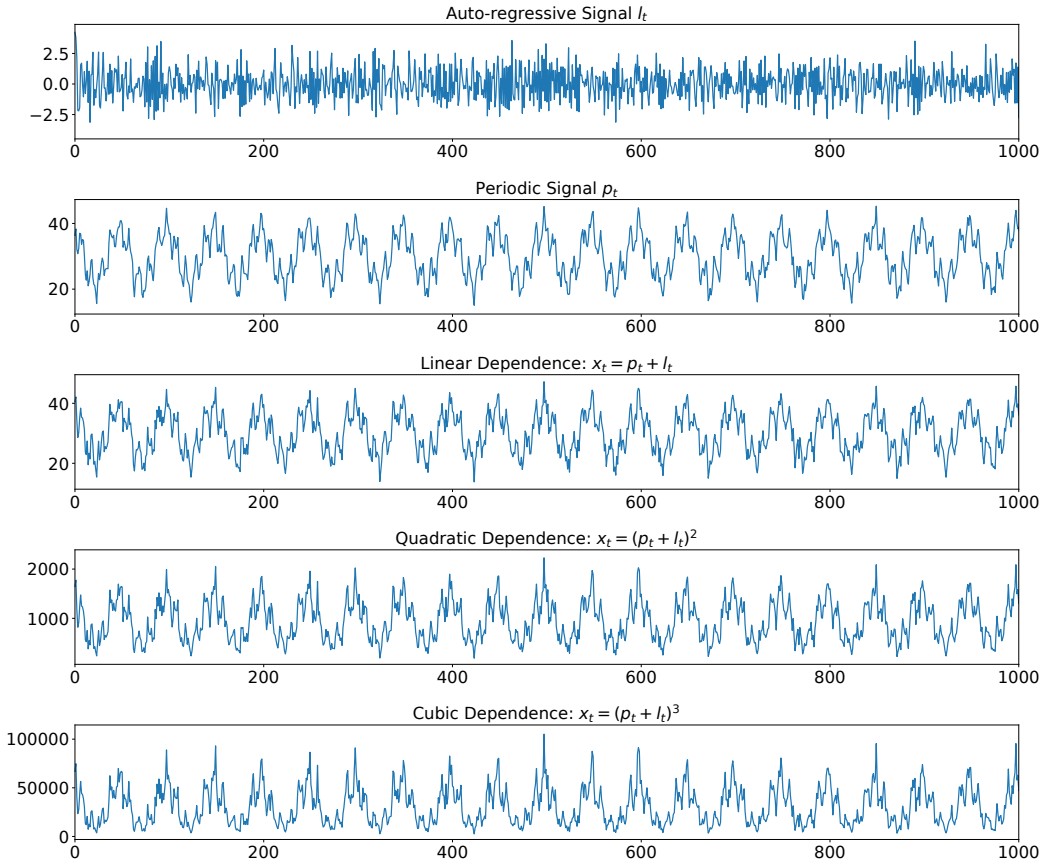

Figure 6: Synthetic Data.

# D   MORE DETAILS ON REAL-WORLD EXPERIMENTS

## D.1   DATASETS

Table 3 includes main statistics of the five datasets used by our experiments. We can see that the existing datasets (ELECTRICITY, TRAFFIC, and M4 (HOURLY)) utilized by recent studies usually have a large number of series but with relatively short lengths. Therefore, it is hard to identify or evaluate yearly or quarterly periods on these benchmarks. In contrast, CAISO and NP contain tens of series with the lengths of several years, which can better illustrate the inherent periodicity of these series and serve as complementary benchmarks for PTS modeling.

Table 3: Dataset statistics.

| Dataset | ELECTRICITY | TRAFFIC | M4 (HOURLY) | CAISO | NP |
|---|---|---|---|---|---|
| # Series | 370 | 963 | 414 | 10 | 18 |
| Frequency | hourly | hourly | hourly | hourly | hourly |
| Start Date | 2012-01-01 | 2008-01-02 | n/a | 2013-01-01 | 2013-01-01 |
| End Date | 2015-01-01 | 2009-03-31 | n/a | 2020-12-31 | 2020-12-31 |
| Min. Length | 4008 | 10560 | 700 | 37272 | 69984 |
| Max. Length | 26304 | 10560 | 960 | 70128 | 70128 |
| Avg. Length | 24556 | 10560 | 854 | 54259 | 70120 |
| Max. Value | 764500 | 1.0000 | 352000 | 49909 | 27513 |
| Avg. Value | 2378.9 | 0.0528 | 1351.6 | 5582.4 | 4671.4 |

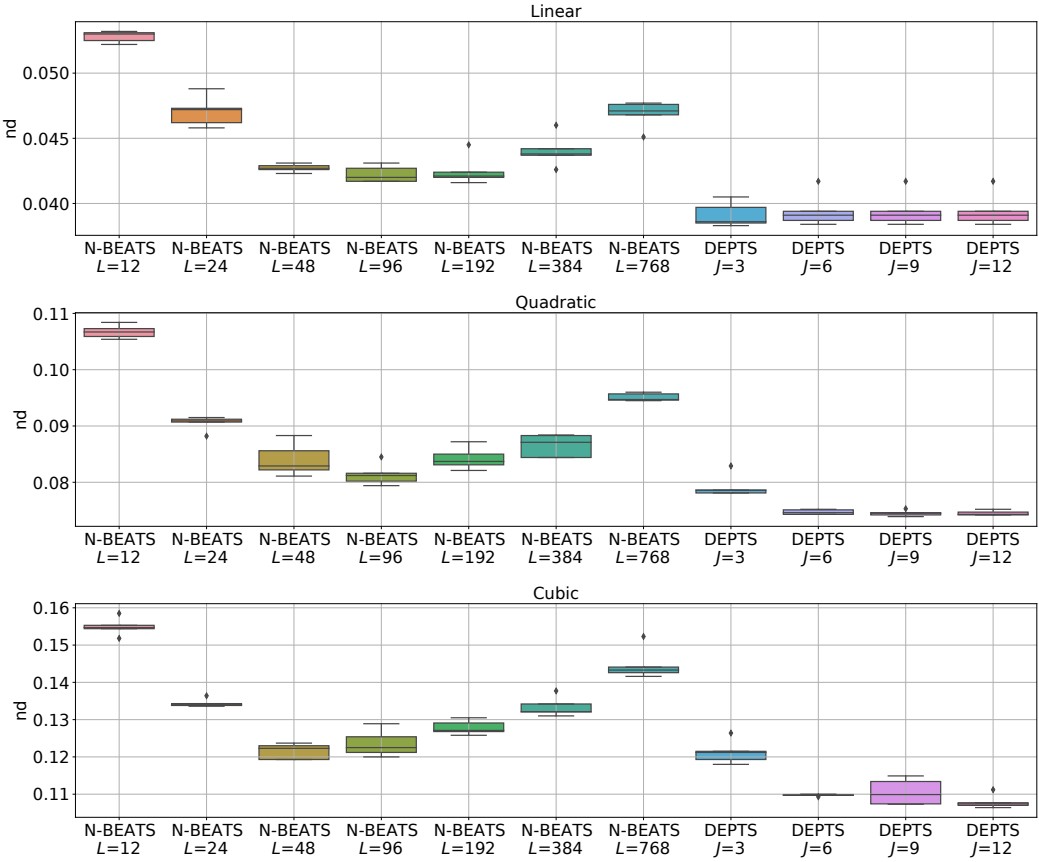

Figure 7: Performance comparisons of N-BEATS and DEPTS with different lookback lengths ($L$) and number of periods ($J$).

Table 4: Hyper-parameters of N-BEATS on CAISO and NP.

| Dataset | CAISO / NP | | | |
|---|---|---|---|---|
| Split | 2020-01-01 | 2020-04-01 | 2020-07-01 | 2020-10-01 |
| Iterations | 4000 / 12000 | | | |
| Loss | sMAPE | | | |
| Forecast horizon ($H$) | 24 | | | |
| Lookback horizon | $2H, 3H, 4H, 5H, 6H, 7H$ | | | |
| Training horizon | $720H$ (most recent points before the split) | | | |
| Layer number | 30 | | | |
| Layer size | 512 | | | |
| Batch size | 1024 | | | |
| Learning rate | 1e-3 / 1e-6 | | | |
| Optimizer | Adam (Kingma & Ba, 2014) | | | |

## D.2 HYPER-PARAMETERS

For N-BEATS, we use its default hyper-parameters[6] for ELECTRICITY, TRAFFIC, and M4 (HOURLY), and we report its hyper-parameters searched on CAISO and NP in Table 4. Besides, N-BEATS used multiple loss functions, such as sMAPE or MASE, for model training, and we also follow these setups. Tables 5 and 6 include the hyper-parameters of DEPTS for all five datasets. Note

---

[6]https://github.com/ElementAI/N-BEATS

Table 5: Hyper-parameters of DEPTS on ELECTRICITY, TRAFFIC, and M4 (HOURLY).

| Dataset | ELECTRICITY | | TRAFFIC | | M4 (HOURLY) |
|---|---|---|---|---|---|
| Split | 2014-09-01 | 2014-12-25 | 2008-06-15 | 2009-03-24 | |
| Iterations | 72000 | | | 12000 | |
| Loss | | sMAPE | | | MASE |
| Forecast horizon ($H$) | | 24 | | | 48 |
| Lookback horizon | | $2H, 3H, 4H, 5H, 6H, 7H$ | | | $4H, 5H, 6H, 7H$ |
| Training horizon | | $10H$ | | | |
| $J$ | 4 | 32 | | 8 | 1 |
| $K$ | | 128 | | | |
| Layer number | | 30 | | | |
| Layer size | | 512 | | | |
| Batch size | | 1024 | | | |
| Learning rate ($f_\theta$) | | 1e-3 | | | |
| Learning rate ($g_\phi$) | | 5e-7 | | | |
| Optimizer | | Adam (Kingma & Ba, 2014) | | | |

Table 6: Hyper-parameters of DEPTS on CAISO and NP.

| Dataset | CAISO / NP | | | |
|---|---|---|---|---|
| Split | 2020-01-01 | 2020-04-01 | 2020-07-01 | 2020-10-01 |
| Iterations | | 4000 / 12000 | | |
| Loss | | sMAPE | | |
| Forecast horizon ($H$) | | 24 | | |
| Lookback horizon | | $2H, 3H, 4H, 5H, 6H, 7H$ | | |
| Training horizon | | $720H$ | | |
| $J$ | 8 / 8 | 32 / 8 | 32 / 32 | 8 / 32 |
| K | | 128 | | |
| Layer number | | 30 | | |
| Layer size | | 512 | | |
| Batch size | | 1024 | | |
| Learning rate ($f_\theta$) | | 1e-3 / 1e-6 | | |
| Learning rate ($g_\phi$) | | 5e-7 | | |
| Optimizer | | Adam (Kingma & Ba, 2014) | | |

that, all these hyper-parameters are searched on a validation set, which is defined as the last week before the test split. Moreover, for a typical dataset with multiple series, we build an independent periodicity module $g_\phi$ for each series and perform respective parameter initialization procedures as described in Appendix B. Then, for all datasets (splits), we train 30 models (6 lookback lengths $\times$ 5 random seeds) for both N-BEATS and DEPTS and then produce ensemble forecasts for fair and robust evaluation.

## E ABLATION TESTS

As Figure 8 shows, we adopt three ablated variants of DEPTS to demonstrate our critical designs in the expansion module (Section 4.2):

- **DEPTS-1**: removing the residual connection of $(\boldsymbol{x}_{t-L:t}^{(\ell-1)} - \boldsymbol{v}_{t-L:t}^{(\ell)})$ so that the outputs of the local block $\boldsymbol{u}_{t-L:t}^{(\ell)}$ are only conditioned on the raw PTS signals $\boldsymbol{x}_{t-L:t}$, which correspond to the mixed observations of local momenta and global periodicity.

- **DEPTS-2**: removing the residual connection of $(\hat{\boldsymbol{x}}_{t:t+H}^{(\ell-1)} + \boldsymbol{v}_{t:t+H}^{(\ell)})$ so that the contributions to the forecasts only come from the local block, which takes in the signals excluding periodic effects progressively.

- **DEPTS-3**: removing the residual connection of $(z_{t-L:t+H}^{(\ell-1)}) - v_{t-L:t+H}^{(\ell)}$ so that the inputs to the periodic block of each layer are the same hidden variables $z_{t-L:t+H}$.

We also construct another four baselines to demonstrate the importance of our customized periodicity learning:

- **NoPeriod**: removing the periodic blocks by directly feeding $(x_t - z_t)$ to N-BEATS.
- **RandInit**: randomly initializing periodic coefficients ($\phi$) and directly applying the end-to-end learning.
- **FixPeriod**: fixing the periodic coefficients ($\phi$) after the initialization stage and only tuning $\theta$ during end-to-end optimization.
- **MultiVar**: treating $z_t$ as a covariate of $x_t$ and feeding $(x_t, z_t)$ into an N-BEATS-style model via two channels.

Moreover, as illustrated in Section 4.3 and Appendix B, the maximal number of selected periods $J$ is a critical hyper-parameter to balance expressiveness and generalization of $g_\phi(t)$. Thus, we conduct experiments with different $J$ to verify its sensitivity on different datasets. Tables 7 and 8 include experimental results of these model variants on ELECTRICITY, TRAFFIC, CAISO, and NP with different $J$. Since we only identify one reliable period via Algorithm 1 on M4 (HOURLY), we report its results separately in Table 9.

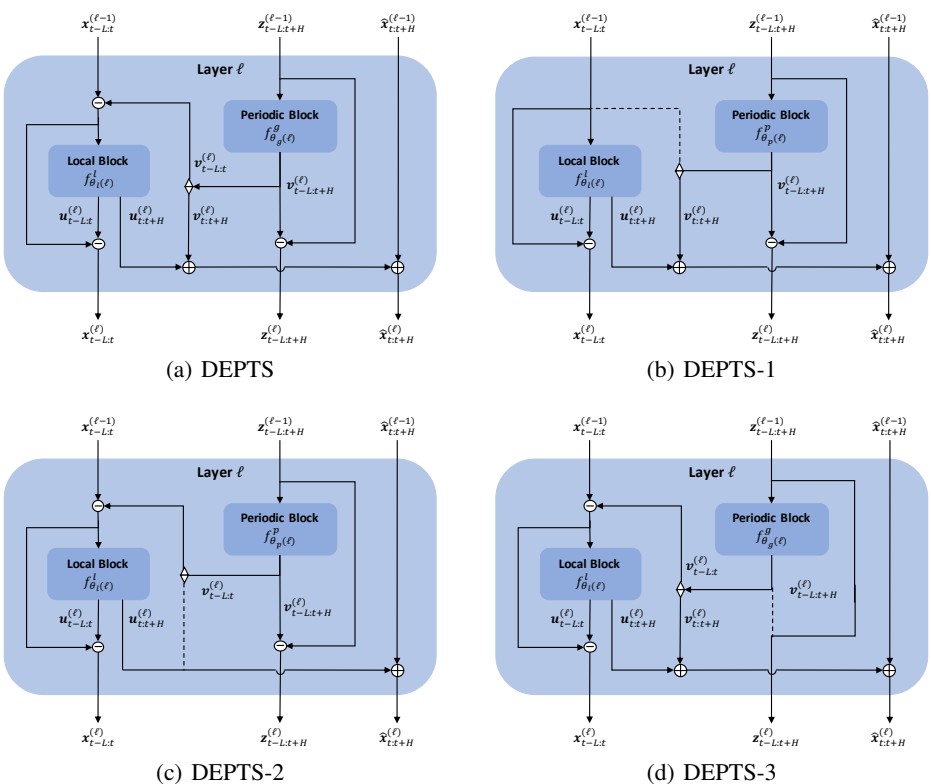

Figure 8: The residual structures of DEPTS and its three ablated variants, where the dashed line denotes the removed connection.

First, as Tables 7 and 8 show, $J$ is a crucial hyper-parameter that has huge impacts on forecasting performance. The reason is that if $J$ is too small, the periodicity module $g_\phi$ cannot produce effective representations of the inherent periodicity to boost the predictive ability. While if it is too large, $g_\phi$ has a high risk of over-fitting to the irrelevant noises contained by the training data, which also results in poor predictive performance. Moreover, the interactions between local momenta and global periodicity may vary over time. Therefore, it is critical to search for a proper $J$ for each PTS and

Table 7: Performance comparisons of DEPTS-1, DEPTS-2, DEPTS-3, and DEPTS.

| $J$ | Model | ELECTRICITY 2014-12-25 | | TRAFFIC 2009-03-24 | | CAISO 2020-10-01 | | NP 2020-10-01 | |
|---|---|---|---|---|---|---|---|---|---|
| | | nd | nrmse | nd | nrmse | nd | nrmse | nd | nrmse |
| 4 | DEPTS-1 | 0.15870 | 0.97571 | 0.11167 | 0.40790 | 0.02429 | 0.05184 | 0.19818 | 0.31224 |
| | DEPTS-2 | 0.15391 | 0.99258 | 0.10786 | **0.39716** | 0.02190 | 0.04402 | 0.19213 | **0.30317** |
| | DEPTS-3 | 0.14955 | 0.96602 | 0.10784 | 0.39811 | **0.01951** | **0.04087** | 0.19201 | 0.30469 |
| | DEPTS | **0.14931** | **0.96488** | **0.10745** | 0.39730 | 0.02061 | 0.04373 | **0.19128** | 0.30381 |
| 8 | DEPTS-1 | 0.15632 | 0.98683 | 0.11108 | 0.40529 | 0.02256 | 0.04634 | 0.19194 | 0.30167 |
| | DEPTS-2 | 0.15070 | 0.97949 | 0.10688 | **0.39421** | 0.02103 | 0.04329 | **0.18412** | **0.28983** |
| | DEPTS-3 | **0.14908** | 0.96270 | 0.10714 | 0.39687 | 0.02017 | 0.04532 | 0.18618 | 0.29525 |
| | DEPTS | 0.14929 | **0.95627** | **0.10653** | 0.39567 | **0.02008** | **0.04176** | 0.18475 | 0.29214 |
| 16 | DEPTS-1 | 0.14954 | 0.92162 | 0.11216 | 0.40335 | 0.02419 | 0.05041 | 0.18740 | 0.29442 |
| | DEPTS-2 | 0.14719 | 0.95648 | 0.10806 | 0.39448 | 0.02236 | 0.04672 | 0.18124 | **0.28434** |
| | DEPTS-3 | 0.14742 | **0.94554** | **0.10678** | 0.39444 | **0.01991** | 0.04384 | 0.18180 | 0.28786 |
| | DEPTS | **0.14653** | 0.94929 | 0.10770 | 0.39554 | 0.02116 | **0.04276** | **0.18095** | 0.28620 |
| 32 | DEPTS-1 | 0.14730 | 0.90305 | 0.11425 | 0.39968 | 0.02476 | 0.05193 | 0.18445 | 0.28860 |
| | DEPTS-2 | 0.14765 | 0.95478 | 0.11061 | 0.39699 | 0.02171 | 0.04526 | 0.18024 | 0.28110 |
| | DEPTS-3 | 0.14179 | 0.90319 | **0.10801** | 0.39403 | **0.01975** | **0.04270** | 0.18057 | 0.28355 |
| | DEPTS | **0.13915** | **0.87498** | 0.11076 | 0.39453 | 0.02156 | 0.04446 | **0.17885** | **0.28031** |

Table 8: Performance comparisons of NoPeriod, RandInit, FixPeriod, MultiVar, and DEPTS.

| $J$ | Model | ELECTRICITY 2014-12-25 | | TRAFFIC 2009-03-24 | | CAISO 2020-10-01 | | NP 2020-10-01 | |
|---|---|---|---|---|---|---|---|---|---|
| | | nd | nrmse | nd | nrmse | nd | nrmse | nd | nrmse |
| 4 | NoPeriod | 0.20615 | 1.27117 | 0.11829 | 0.40465 | 0.08195 | 0.14208 | 0.27128 | 0.40491 |
| | RandInit | 0.17677 | 1.07514 | 0.11051 | 0.40383 | 0.02504 | 0.05293 | 0.20869 | 0.32853 |
| | FixPeriod | 0.16756 | 0.99876 | 0.10816 | 0.39833 | 0.02282 | 0.04576 | 0.20145 | 0.31604 |
| | MultiVar | 0.15743 | 1.02039 | **0.10733** | **0.39635** | **0.02018** | **0.04038** | 0.19998 | 0.31393 |
| | DEPTS | **0.14931** | **0.96488** | 0.10745 | 0.39730 | 0.02061 | 0.04373 | **0.19128** | **0.30381** |
| 8 | NoPeriod | 0.23969 | 1.47537 | 0.11940 | 0.40536 | 0.08182 | 0.15585 | 0.24796 | 0.37781 |
| | RandInit | 0.17463 | 1.05695 | 0.11065 | 0.40500 | 0.02639 | 0.05680 | 0.20972 | 0.33044 |
| | FixPeriod | 0.16431 | 0.98414 | 0.10796 | 0.39764 | 0.02228 | 0.04661 | 0.20163 | 0.31666 |
| | MultiVar | 0.15482 | 0.99266 | 0.10667 | 0.39599 | 0.02160 | 0.04855 | 0.19494 | 0.30513 |
| | DEPTS | **0.14929** | **0.95627** | **0.10653** | **0.39567** | **0.02008** | **0.04176** | **0.18475** | **0.29214** |
| 16 | NoPeriod | 0.26851 | 1.65571 | 0.12203 | 0.40410 | 0.07275 | 0.15280 | 0.23281 | 0.34943 |
| | RandInit | 0.18167 | 1.08529 | 0.11048 | 0.40287 | 0.02496 | 0.05347 | 0.20917 | 0.32936 |
| | FixPeriod | 0.15792 | 0.95356 | 0.10803 | 0.39699 | 0.02138 | 0.04417 | 0.20293 | 0.31963 |
| | MultiVar | 0.15479 | 0.98805 | **0.10724** | **0.39366** | 0.02289 | 0.05023 | 0.19045 | 0.29784 |
| | DEPTS | **0.14653** | **0.94929** | 0.10770 | 0.39554 | **0.02116** | **0.04276** | **0.18095** | **0.28620** |
| 32 | NoPeriod | 0.31358 | 1.73706 | 0.12835 | 0.40741 | 0.07696 | 0.15433 | 0.22503 | 0.34109 |
| | RandInit | 0.19399 | 1.14278 | 0.11075 | 0.40082 | 0.02577 | 0.05723 | 0.20916 | 0.32944 |
| | FixPeriod | 0.15539 | 0.92896 | **0.10862** | 0.39637 | **0.02055** | **0.04115** | 0.20272 | 0.31940 |
| | MultiVar | 0.16405 | 1.01227 | 0.10907 | 0.39596 | 0.02159 | 0.04582 | 0.18844 | 0.29294 |
| | DEPTS | **0.13915** | **0.87498** | 0.11076 | **0.39453** | 0.02156 | 0.04446 | **0.17885** | **0.28031** |

each split point to pursue better performance. Fortunately, we demonstrate that the hyper-parameter tuning of $J$ on the validation set can ensure its good generalization abilities on the subsequent test horizons.

Then, let us focus on Table 7 to compare DEPTS with DEPTS-1, DEPTS-2, and DEPTS-3. First, we can see that DEPTS-1 usually produces the worst performance in most cases, which demon-

Table 9: Overall ablation studies on M4 (HOURLY).

|  | DEPTS-1 | DEPTS-2 | DEPTS-3 | DEPTS |
|---|---|---|---|---|
| *nd* | 0.03712 | 0.02161 | 0.02252 | **0.02050** |
| *nrmse* | 0.22785 | 0.07710 | 0.08851 | **0.06872** |
|  | NoPeriod | RandInit | FixPeriod | MultiVar |
| *nd* | 0.03848 | 0.02401 | 0.02526 | 0.02937 |
| *nrmse* | 0.16568 | 0.09192 | 0.11221 | 0.14130 |

strates that excluding periodic effects from raw PTS signals can stably and significantly boost the performance of PTS forecasting. Second, for most cases, DEPTS outperforms DEPTS-2, and the performance gaps can be remarkable, such as $0.139$ vs. $0.148$ on ELECTRICITY and $0.020$ vs. $0.021$ on CAISO. These results verify the importance of including the portion of forecasts solely from the periodicity module. Third, DEPTS-3 can produce competitive results compared with DEPTS in many cases. Nevertheless, after selecting the best $J$ for each dataset, DEPTS still slightly outperforms DEPTS-3 in most cases. Besides, from Table 9, we also observe that DEPTS performs much better than DEPTS-3 on M4 (HOURLY). Thus we retain the residual connection to reduce the periodic effects leveraged by previous layers.

Next, let us focus on Table 8 to compare DEPTS with other four baselines, NoPeriod, RandInit, MultiVar, and FixPeriod. First, we observe that NoPeriod usually produces the worst preformance. The reason is that $(x_t-z_t)$ denotes the raw time-series signal subtracting the periodic effect, so it is challenging for the model to forecast the future signals, $x_{t:t+H}$, solely based on the periodicity-agnostic inputs, $(x_{t-L:t}-z_{t-L:t})$. Second, RandInit also produces much more worse results than DEPTS, which demonstrate the importance of initializing periodic coefficients (Section 4.3). Third, DEPTS performs much better than FixPeriod in most cases, which demonstrate the effectiveness of fine-tuning periodic coefficients after the initialization stage. Last, we observe that sometimes MultiVar can produce comparable and even slightly better results than DEPTS. However, after selecting the best $J$ on each dataset for these two models, we find that DEPTS still outperforms MultiVar consistently and significantly, which also demonstrates the superiority of our expansion learning. Moreover, as Table 9 shows, DEPTS outperforms all these baselines by a large margin on M4 (HOURLY), which containing very short PTS with 854 observations on average. Given limited data, all our critical designs, such as properly initializing periodic coefficients, fine-tuning periodic coefficients, and conducting expansion learning to decouple the dependencies of $x_t$ on $z_t$, play much crucial roles in producing accurate forecasts.

## F MORE CASE STUDIES AND INTERPRETABILITY ANALYSES

In the following, we further study the interpretable effects of DEPTS with more cases. Figures 9, 10, 11 and 12 show the additional two cases on ELECTRICITY, TRAFFIC, CAISO, and NP, respectively. Following Figure 4, we compare the forecasts of N-BEATS and DEPTS in the left side, differentiate the forecasts of DEPTS into the local part (DEPTS-L) and the periodic part (DEPTS-P) in the middle side, and plot the hidden state $z_t$ together with the PTS signals in the right side. The general observations are that with the help of explicit periodicity modeling, DEPTS achieves better performance than N-BEATS in PTS forecasting, and DEPTS has learned diverse behaviors for different cases. Besides, we also include their critical periodic coefficients (amplitude $A_k$, frequency $F_k$, and phase $P_k$) in Tables 10, 11, 12, and 13. We find that DEPTS can learn many meaningful periods that are consistent with practical domains.

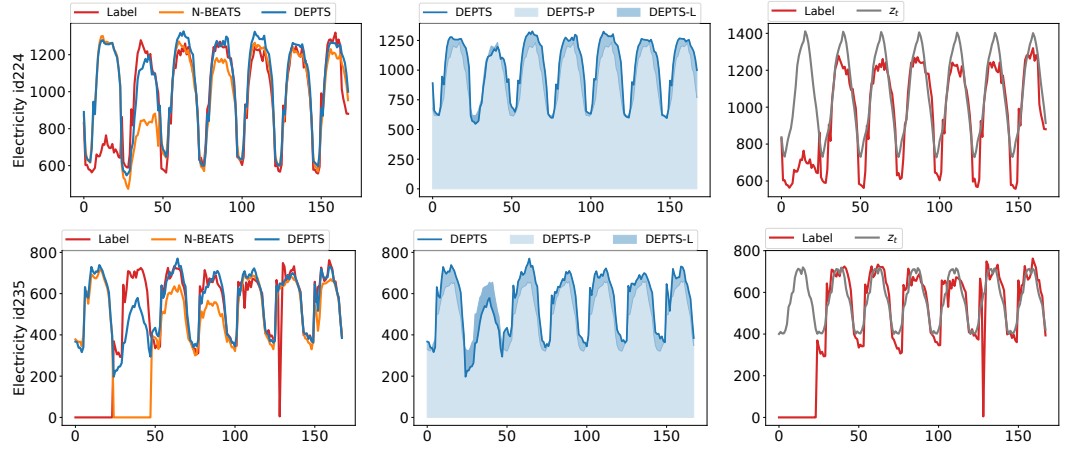

Figure 9: We show two cases on ELECTRICITY dataset. It is clear to see that other than following some inherent periodicity, the real PTS signals usually have various irregular oscillations at different time steps, while DEPTS can produce more stable forecasts by analyzing local momenta and global periodicity simultaneously. For these two cases with evident and stable periodicity, DEPTS relies more on the periodic forecasts (DEPTS-P) and thus achieves more competitive and stable results.

Table 10: Periodic coefficients of the two ELECTRICITY examples shown in Figure 9. We find that DEPTS has learned both short-term and long-term periods, such as three hours ($|1/F_k| \approx 3$), six hours ($|1/F_k| \approx 6$), 12 hours ($|1/F_k| \approx 12$), one day ($|1/F_k| \approx 24$), and half a year ($|1/F_k| \approx 4380$), which are very similar to the patterns of electricity utilization in practice.

| ELECTRICITY | | | | | |
|---|---|---|---|---|---|
| id 224 | | | id 235 | | |
| $|A_k|$ | $|1/F_k|$ | $|P_k|$ | $|A_k|$ | $|1/F_k|$ | $|P_k|$ |
| 362.601 | 23.995 | 0.088 | 160.144 | 23.997 | 0.092 |
| 196.829 | 8320.428 | 0.422 | 77.804 | 8256.523 | 0.451 |
| 87.138 | 4470.598 | 0.487 | 36.918 | 23.969 | 0.092 |
| 66.418 | 24.035 | 0.122 | 19.517 | 23.921 | 0.102 |
| 52.736 | 11.999 | 0.052 | 17.714 | 4.800 | 0.024 |
| 44.248 | 23.920 | 0.096 | 12.810 | 11.993 | 0.054 |
| 27.172 | 6.000 | 0.027 | 11.964 | 6068.298 | 0.684 |
| 23.220 | 6.001 | 0.030 | 11.186 | 3.000 | 0.015 |

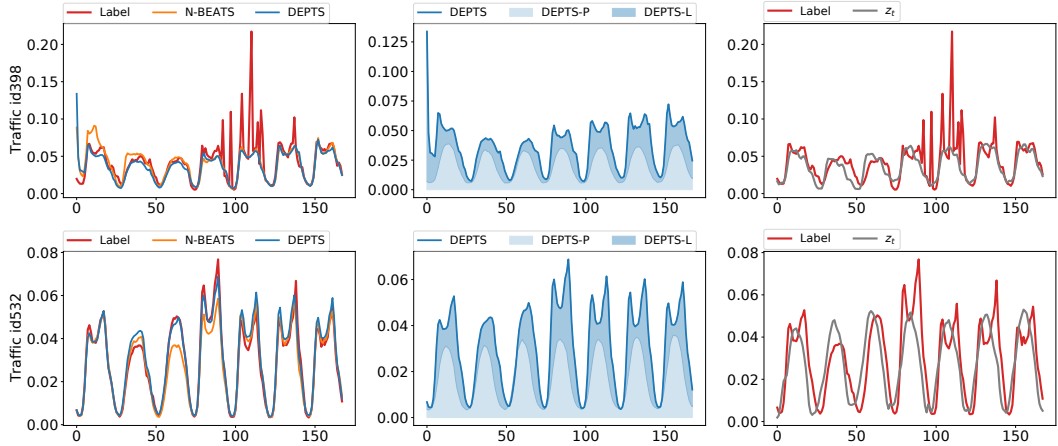

Figure 10: We show two cases on TRAFFIC dataset. We can see that DEPTS is able to characterize quite different periodic effects. For the upper case, there are unexpected peaks at different time steps. For the bottom case, there are different types of periodic oscillations. Similar to cases in Figure 9, DEPTS has estimated roughly consistent periodic states $z_t$ and then combined DEPTS-P and DEPTS-L to produce stable and accurate forecasts.

Table 11: Periodic coefficients of the two TRAFFIC examples shown in Figure 10. We find that DEPTS has also learned multiple types of periods.

| | TRAFFIC | | | | |
| --- | --- | --- | --- | --- | --- |
| | id 398 | | | id 532 | |
| $|A_k|$ | $|1/F_k|$ | $|P_k|$ | $|A_k|$ | $|1/F_k|$ | $|P_k|$ |
| 0.0231 | 23.993 | 0.233 | 0.0267 | 23.987 | 0.209 |
| 0.0066 | 164.055 | 2.293 | 0.0122 | 3192.218 | 0.527 |
| 0.0062 | 1845.505 | 2.226 | 0.0104 | 4906.324 | 0.382 |
| 0.0054 | 11.999 | 0.200 | 0.0066 | 24.270 | 0.434 |
| 0.0046 | 8.003 | 0.205 | 0.0048 | 1265.645 | 0.330 |
| 0.0045 | 23.920 | 0.234 | 0.0039 | 4.801 | 0.114 |
| 0.0044 | 28.097 | 0.282 | 0.0032 | 23.637 | 0.332 |
| 0.0038 | 12.011 | 0.513 | 0.0030 | 28.053 | 0.248 |

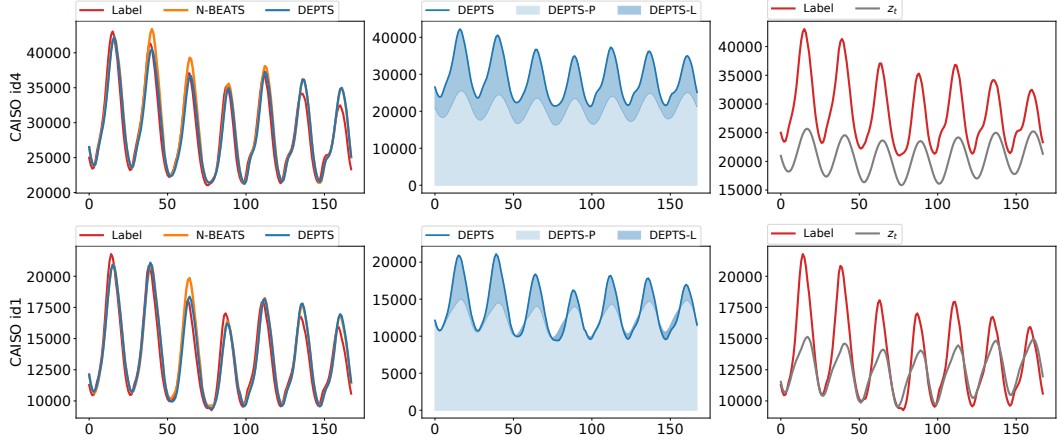

Figure 11: We show two cases on CAISO dataset. These two cases present relatively regular oscillations, and thus N-BEATS with enough lookback lengths can also produce pretty good forecasts. Even though, DEPTS can better capture the curves of future PTS signals by modeling the dependencies of them on estimated periodicity. We can see that DEPTS first relies on the periodic part (DEPTS-P) to form the basic shape of forecasts and then leverages the forecasts from the local part (DEPTS-L) to stretch or condense the forecasting curve.

Table 12: Periodic coefficients of the two CAISO examples shown in Figure 11. Other than daily and yearly periods, which are observed similarly in ELECTRICITY and TRAFFIC cases, we find that DEPTS has identified some weekly periods ($|1/F_k| \approx 168$) for two cases on CAISO.

|  | | | | | |
|---|---|---|---|---|---|
| | | | CAISO | | |
| | id 4 | | | id 1 | |
| $|A_k|$ | $|1/F_k|$ | $|P_k|$ | $|A_k|$ | $|1/F_k|$ | $|P_k|$ |
| 3411.731 | 24.004 | 0.000 | 1851.303 | 24.002 | 0.307 |
| 3207.279 | 8344.825 | 0.081 | 1754.576 | 8629.984 | 0.606 |
| 1712.536 | 4509.138 | 0.042 | 720.007 | 23.934 | 0.704 |
| 1493.276 | 23.934 | 0.000 | 625.465 | 4299.993 | 0.334 |
| 1434.023 | 23.992 | 0.000 | 536.312 | 167.907 | 0.369 |
| 1225.926 | 9408.412 | 0.086 | 452.345 | 11.999 | 0.086 |
| 963.309 | 24.062 | 0.000 | 409.348 | 24.018 | 0.323 |
| 854.321 | 168.236 | 0.001 | 326.875 | 24.069 | 0.215 |

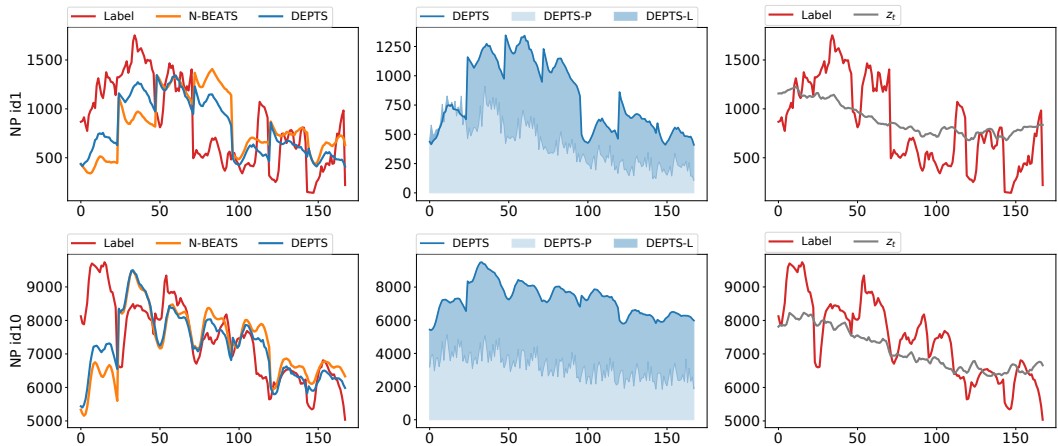

Figure 12: We show two cases on NP dataset. We can see that these cases are rather difficult, and both N-BEATS and DEPTS struggle to make sufficiently accurate forecasts. Nevertheless, as shown in the right side, DEPTS has a relatively stable estimation of the future trending and thus can obtain relatively good performance in forecasting future curves.

Table 13: Periodic coefficients of the two NP examples shown in Figure 12. We can see that the dominant periods belong to the long-term type, which characterizes the overall variation but omits those local volatile oscillations. Since this dataset contains massive noises in local oscillations, in some splits, N-BEATS even produces forecasts that are inferior to the projections of simple statistical approaches, such as PARMA, as shown in Table 2.

| NP | | | | | |
|---|---|---|---|---|---|
| id 1 | | | | id 10 | |
| $\|A_k\|$ | $\|1/F_k\|$ | $\|P_k\|$ | $\|A_k\|$ | $\|1/F_k\|$ | $\|P_k\|$ |
| 252.601 | 8529.557 | 0.960 | 2131.096 | 8506.317 | 0.809 |
| 140.967 | 670.924 | 0.715 | 1643.707 | 24.003 | 0.212 |
| 134.343 | 366.735 | 0.668 | 1158.437 | 366.648 | 0.797 |
| 117.755 | 24.004 | 0.378 | 1132.171 | 670.602 | 0.734 |
| 107.298 | 244.195 | 0.768 | 942.847 | 244.106 | 0.988 |
| 97.824 | 794.904 | 0.376 | 909.762 | 795.685 | 0.417 |
| 69.710 | 6217.354 | 0.746 | 867.654 | 12.001 | 0.309 |
| 65.251 | 182.112 | 0.452 | 729.163 | 737.857 | 0.696 |

