# OpenReview forum: "DEPTS: Deep Expansion Learning for Periodic Time Series Forecasting"
_ICLR.cc/2022/Conference — ICLR 2022 Spotlight_

### Official Review · Reviewer_qo99 · 2021-11-02

**Correctness:** 3
**Technical Novelty And Significance:** 3
**Empirical Novelty And Significance:** 3
**Recommendation:** 8
**Confidence:** 3

**Main Review:**

### strong points
1. a customized deep learning (DL) architecture for periodic time series to explicitly take account of the periodic property
2. the proposed techniques are reasonable and sound novel.
3. extensive experiments on a synthetic dataset and several real-world datasets about the effectiveness and interpretability.

### weak points
1. The writing of the paper should be improved.
2. Some technical details are missing and some motivations are not very clear.
3. lack of significant test
4. lack of complexity analysis

**Summary Of The Paper:**

This paper proposed a novel DL framework for periodic time series forecasting. The contributions of the paper include modeling complicated periodic dependency and capturing compositions of diversified periods. The authors conduct extensive experiments on synthetic data and real-world datasets, showing the effectiveness and interpretability of the proposed methods

**Summary Of The Review:**

In this paper, the authors proposed a customized deep learning (DL) architecture for periodic time series to explicitly take account of the periodic property. The proposed solution is technically sound and effective based on the extensive evaluation. The periodical module sounds novel and reasonable, and parameter initialization resorts to a two-stage optimization problem, which is also interesting.  However, I have the following comments that the authors should take into account to improve the paper.

1. the writing should be improved. For example, Section 4.2 is not easy to read and the left part of Figure 2 is not very easy to understand. The authors are suggested to overview the framework before delving into the detailed description. The figure should be made more clear to understand.
2. a few technical parts are missing. For example, how to design local block function f^l and periodic block function f^p.
3. some motivations are not very clear. For example, why instantiate the period function as a series of cosine functions and how such a formulation can lead to the modeling of periodic patterns.
4. Though the results of the proposed method are good, but in some cases, the improvements are small. The authors are suggested to include significant testing and show some standard error in the result table.
5.  Though the proposed method performs well, it is unclear the efficiency of the proposed method compared to baselines. The authors are suggested to add detailed complexity analysis to the proposed methods and some related baselines.

---

> ### Author Response · Authors · 2021-11-16
> **Response**
>
> Thanks very much for your review. We really appreciate many of your constructive suggestions on improving the paper.
>
> 1. We are so sorry that the current writing is not clear enough for you to understand. We will follow your advice to provide more motivations and the overall picture in Section 4.2. Besides, we plan to add a hint to encourage audiences to align Eq. 4 with the left part of Figure 2, which may facilitate better understanding. Please let us know if you have better ideas on improving this part.
> 2. Please check Appendix A for block architectures.
> 3. In general, we need a periodic function, g(t), to obtain the specific periodic state, z_t, for a timestamp, t, as z_t = g(t). Any periodic functions are allowed in this framework. In our instantiation, we adopt the composition of cosine functions due to two considerations:
>     - It is simple and effective. Compared with using Fourier series, using cosine series does not involve complex numbers. Even with such a simple periodic representation function, we demonstrate that it is effective as it can boost the performance of SOTA deep learning models significantly.
>     - It shares a similar format with the Discrete Cosine Transform. This connection can help us to tackle the difficulties in the optimization of Eq. 5, as discussed in Section 4.3 and Appendix B.
> 4. We totally agree with you that including significant testing and showing some standard errors can improve the soundness of experiments and consolidate our conclusions. For the synthetic experiments, we already plot the error distributions, which demonstrate that the performance gaps are statistically significant. For real-world experiments , we follow previous studies, such as N-BEATS, to produce the ensemble forecasts (30 models for each dataset split, 5 random seeds x 6 lookback horizons). In this way, the resulting performance has very small variances. Here we also provide the 95% confidence intervals of experimental results for your reference.
> |Dataset|N-BEATS|DEPTS|
> | :------ | ------: | ------:|
> |Electricity (2014-12-15)| 0.171±0.00527|0.139±0.00404|
> |Traffic (2009-03-24)| 0.112±0.00349|0.107±0.00112|
> |M4 (Hourly) | 0.023±0.00136|0.021±0.00124|
> |CAISO (2020-10-01)| 0.026±0.00074|0.020±0.00067|
> |NP (2020-10-01)| 0.211±0.00104|0.179±0.00169|
> 5. Compared with all baselines, DEPTS introduces a new initialization stage for the periodicity module. In effect, our approximation algorithm for Eq. 5 can run very fast. Here we supplement the complexity of this approximation algorithm as you suggested. Assume the length of training series is $L_t$, and the length of validation series is $L_v$. The complexity of conducting DCT over training series is $O(L_t log(L_t))$. Then, the complexity of selecting top-K frequencies with the largest amplitudes is $O(L_t log(K))$, which can be ignored since $K << L_t$. Next, we need to select at most $J$ frequencies greedily based on the generalization errors on the validation set. Since we measure the generalization errors via dynamic time warping, the total worst complexity for this selection procedure is $O(K L_v^2)$. In total, the worst complexity of our approximation algorithm for a series is $O(L_t log(L_t) + K L_v^2)$. In practice, $L_v$, the length of the validation series, is relatively small, and $K$, the maximum number of frequencies, can be regarded as a small constant. So, the squared complexity term $O(K L_v^2)$ is not a big trouble. According to our experiences, the time spent on the initialization stage is negligible compared with the time spent on training neural networks. At last, thanks for this suggestion, we will include this complexity analyses into the appendix of the revised paper.

---

### Official Review · Reviewer_xtYj · 2021-11-02

**Correctness:** 3
**Technical Novelty And Significance:** 3
**Empirical Novelty And Significance:** 3
**Recommendation:** 8
**Confidence:** 4

**Main Review:**

The paper is very well written and easy to follow. The proposed
model is plausible and the experiments show a lift. The paper is
accompagnied by a detailed appendix with further experiments,
esp. ablation studies. Methodologically, the paper is a
small extension of N-BEATS, but due to modelling the periodicity
explicitly an interesting one.

However, two questions about the proposed model are not
answered in the paper currently:

1. Does learning the periodic time encoding g_phi and the forecasting
   model f_theta really have advantages? One could imagine a simple
   baseline, where the learnt periodic time encoding is fed as covariate
   channel into N-BEATS. Is this as good as the proposed model?
   -- Also an ablation study in which the time encoder g_phi is frozen
   after the first stage, could shed some light on this question.

2. It is not clear on what hyperparameter grid the hyperparameter
   optimization has been conducted. Esp. in table 4 and 6, for N-BEATS
   and the proposed model the same backbone architecture seems to
   be optimal, while the proposed model gets additional complexity
   through the time encoder. Have N-BEATS architectures with a similar
   number of parameters as the optimal architecture for the proposed
   model been inside the hyperparameter grid for N-BEATS?


Small questions:
- what is the role of the two-stage optimization problem in eq. 5? If
  I understand it correctly, this problem never is tackled, but the
  authors use the heuristic fitting outlined in the two bullet points
  below eq. 5. Why do we need eq. 5 then?


**Summary Of The Paper:**

The paper addresses the problem of time series forecasting, esp.
with periodic dependencies. The authors propose a model that combines
a learnt one-dimensional sum of cosines as periodic signal with
residual feedforward neural network (N-BEATS). They propose to
learn the model in two stages: first to estimate the cosines
with a discrete cosine transform and greedily selecting the
K ones with largest amplitude, second to refine both the
periodic time encoding and the overall forecasting model
end-to-end. In experiments on synthetic, three real-life datasets
from the literature and two new real-life datasets with longer
training segments, that should allow to identify periodicities
more easily, they show that they outperform the underlying
N-BEATS model mostly consistently.


**Summary Of The Review:**

A well written paper with a small, but interesting contribution.
Some aspects of the evaluation still can be made more clear.

---

> ### Author Response · Authors · 2021-11-16
> **Response (Part 1/2)**
>
> Thanks very much for your review. We would like to answer your insightful questions to address the remaining concerns.
>
> Question (1): “Does learning the periodic time encoding g_phi and the forecasting model f_theta really have advantages? One could imagine a simple baseline, where the learnt periodic time encoding is fed as covariate channel into N-BEATS. Is this as good as the proposed model? -- Also an ablation study in which the time encoder g_phi is frozen after the first stage, could shed some light on this question.”
>
> Yes, our two dedicatedly designed modules, $g_\phi$ and $f_\theta$, are critical to the performance improvements on PTS forecasting. To provide concrete evidences for this claim, we conduct additional studies as you suggested.
>
> The first model variant, denoted as FixPeriod, is to reuse DEPTS but freeze $g_\phi$ after the initialization stage. Moreover, you have mentioned another simple baseline that treats $z_t$ as a covariate of $x_t$ and feeds them together into N-BEATS. While this setup changes the input of N-BEATS from univariate to multivariate and requires some modifications to the initial architecture, we also implement this model variant, denoted as MultiVar. Below we include the comparison results of N-BEATS, FixPeriod, MultiVar, and DEPTS.
>
> | Model |   Electricity (2014-12-15) |   Traffic (2009-03-24) |   CAISO (2020-10-01) |   NP (2020-10-01) | M4 (Hourly) |
> | :------ | ------: | ------: | ------: | ------: | ------: |
> | N-BEATS | 0.17100 | 0.11200 | 0.02553 | 0.21130| 0.02300 |
> | FixPeriod   |    0.15539 |   0.10796 | 0.02055 | 0.20145 | 0.02526|
> | MultiVar |    0.15479 |   0.10667 | 0.02018 | 0.18844 | 0.02937|
> | DEPTS   |    0.13915 |   0.10653 | 0.02008 | 0.17885 | 0.02050|
>
> From the table above, it is obvious to see the performance improvement of DEPTS over FixPeriod, which is mainly attributed to the step of fine-tuning periodic coefficients. The underlying reason is that the initialized periodic coefficients are only rough approximations, and especially, the errors introduced by the periodic module have direct influence on all layers of the expansion module, which hurt the forecasting performance.
>
> Comparing MultiVar with N-BEATS and DEPTS on Electricity, Traffic, CAISO, and NP benchmarks, we can see that simply introducing the periodic effects into a multivariate N-BEATS variant can bring certain performance gains over the vanilla N-BEATS. Nevertheless, DEPTS can still obtain better performance than MultiVar, which demonstrates that conducting the expansion learning may more effectively characterize the real-world data generation procedures.
>
> Moreover, we have an exceptionally different observation on M4 (Hourly) dataset: both FixPeriod and MultiVar perform much worser than N-BEATS, but DEPTS can still obtain 10% improvements over that. We think the reason is related to the ‘short-in-length’ property of M4 (Hourly). Its series only have hundreds of observations, but other datasets have tens of thousands of data points per series (see Table 3 in Appendix D.1). This property greatly increases the difficulties in initializing and optimizing periodic coefficients. With the help of residual expansions, DEPTS achieves much smoother gradient backpropagations from labels to periodic coefficients and thus enables more efficient optimization given limited data. In contrast, FixPeriod loses the chances to further rectify periodic coefficients, and MultiVar suffers more from limited data.
>
> At last, we would like to mention another unique advantage of DEPTS over MultiVar: the interpretability effect. The residual expansions of DEPTS can distinguish the contributions to forecasts into local momenta and global periodicity explicitly, which inform us how the inherent periodicity dynamically influences real observations over time.

---

> > ### Author Response · Authors · 2021-11-16
> > **Response (Part 2/2)**
> >
> > Question (2): “It is not clear on what hyperparameter grid the hyperparameter optimization has been conducted. Esp. in table 4 and 6, for N-BEATS and the proposed model the same backbone architecture seems to be optimal, while the proposed model gets additional complexity through the time encoder. Have N-BEATS architectures with a similar number of parameters as the optimal architecture for the proposed model been inside the hyperparameter grid for N-BEATS?”
> >
> > We agree with you that the proposed model gets additional complexity than N-BEATS, but we also verify that increasing the capacity of N-BEATS (enlarge the hidden dimension or the number of layers) on all five datasets does not lead to statistically consistent improvements. More importantly, this limited complexity introduced by DEPTS enables a new capability of periodicity modeling and contributes consistent and significant improvements.
> >
> > To be more specific, we conduct extensive searching of hyper-parameters about both the training process (training iterations in [4000, 8000, 12000, 36000, 72000], learning rates in [1e-3, 1e-4, 1e-5, 1e-6], training horizons in [10H, 60H, 180H, 360H, 720H]) and the model capacity (number of layers in [20, 30, 40], layer size in [256, 512]) for N-BEATS. On Electricity, Traffic, and M4 (Hourly) benchmarks, we do not find other configurations that robustly beat the initial setup reported by N-BEATS, so we reuse the initial configurations reported by N-BEATS. While for CAISO and NP datasets, we report the hyper-parameters for N-BEATS in Table 4, which are already the best setups we can find to guarantee sufficient capacities for data modeling and to ensure efficient optimization and generalization.
> >
> > Moreover, our ablation tests in Appendix E also shed some lights on this question. Since all three model variants exactly have the same number of parameters as DEPTS, but DEPTS still produces obvious improvements over them, which demonstrate that it is our dedicated design but not the model capacity that plays a critical role in these performance gaps.
> >
> >
> > Questions (3): “what is the role of the two-stage optimization problem in eq. 5? If I understand it correctly, this problem never is tackled, but the authors use the heuristic fitting outlined in the two bullet points below eq. 5. Why do we need eq. 5 then?”
> >
> > The role of the two-stage optimization problem is to present the conditions that identify proper periodic coefficients. We demonstrate the precise mathematical form of this two-stage optimization problem using Eq.5. Nevertheless, since exactly optimizing this problem is very challenging and time-consuming, we derive an approximation method, which is briefly illustrated in the bullet points below Eq. 5 and is elaborated in Appendix B (see Algorithm 1). We believe Eq. 5 is also quite valuable to present audiences with sufficient motivations and insights on how we make a tradeoff and develop an approximation algorithm.

---

### Official Review · Reviewer_3Zt1 · 2021-11-03

**Correctness:** 4
**Technical Novelty And Significance:** 4
**Empirical Novelty And Significance:** 4
**Recommendation:** 8
**Confidence:** 4

**Main Review:**

This paper brings us more attention to the periodicity modeling of time series forecasting in deep learning scenarios. After carefully checking the experiments on synthetic data (Figure 3 and Figure 7), I have noticed that even adding simple operations and periods on the auto-regressive signal data can lead to the state-of-the-art model (N-BEATS) degrade. This is quite meaningful and signifies the importance of periodicity on pure DL-based forecasting models. I think such simple yet intuitive observation can bring up good insights.

The paper has proposed a new decoupled formulation of time series forecasting and formulate the forecasting problem in the model DEPTS by introducing a prior estimated hidden periodic state and a expansion framework based on residue learning. Actually, this formulation is very interesting because most existing works are only to sample series based on the time window for training, which naturally fail to consider given series as whole and thus fail to capture all the information of a series globally. The proposed formulation cleverly makes use of periodicity to relieve this problem by the hidden periodic state, as those low-frequency periods need such modeling in particular. DEPTS framework builds upon the residual learning and expand the learned local residuals and periodic residuals block by block. The periodic blocks and local blocks process the periodicity and the local momenta respectively through expansion. In brief, the whole architecture design is straightforward and it can also be interpretable by outputs of different blocks.

This paper seems to have solid, extensive and diverse experiments. The extensive experimental results have clearly demonstrated the paper intuition (synthetic experiments), the performance improvement (real-world experiments) and the interpretability. The real-world experiments are on 5 different datasets, in which they follow settings of previous work on 3 datasets and construct 2 new datasets. The proposed model achieves better performances across all the datasets. Especially, I find out that DEPTS has a 10% improvement for M4 competition compared with state-of-the-art N-BEATS, which could very challenging. The sufficient interpretability cases are showed in each dataset. Overall, the experiment part is convincing.

Also, I do have some concerns. My first concern is about periodic module. As the paper states, the expansion module and the periodicity module jointly work for the forecasting based on many stacked layers. Each layer includes a local block and a periodic block for the residual expansion. The periodicity module is started with a parameter initialization, which is firstly initialized on the observation data. So my question is if the data is with an obvious trend (e.g., upward trend), how does the periodicity work since it can only handle periods with accumulated cosine functions? My another concern is with the training of the periodic module. If the periodic module is randomly initialized, can this DEPTS model still perform well? Would it still can accurately capture the obvious periods? I am very curious about the forecasting results without initialization of the periodicity module.

**Summary Of The Paper:**

This paper focuses on an important property of time series, periodicity, and mainly studies the problem of periodic time series (PTS) forecasting. This work solves two main research challenges of PTS forecasting: (1) to learn the dependencies of observation data on periodicity; (2) to learn the sophisticated compositions of various periods. The authors propose a deep expansion framework on top of residue learning for dependency learning and a parameterized surrogate function for periodicity learning. Extensive experiments on both synthetic data and 5 real-world datasets demonstrate a significant improvement on time series forecasting tasks when considering periodicity especially.

**Summary Of The Review:**

The paper solves the periodic time series forecasting problem; the proposed DEPTS model is intuitive, effective and interpretable; the experiment is solid and extensive; the improvement is significant.

---

> ### Author Response · Authors · 2021-11-16
> **Response**
>
> Thanks very much for your review. We totally agree with you that the time-series periodicity modeling in deep learning scenarios should receive more research attention, since this property is pervasive in a wide range of real-world applications. We hope the following clarifications can precisely address your concerns.
>
> Question (1): “So my question is if the data is with an obvious trend (e.g., upward trend), how does the periodicity work since it can only handle periods with accumulated cosine functions?”
>
> The periodic blocks of DEPTS do not take over the trend modeling. Instead, we leave this functionality to the local blocks, which take in raw time-series signals excluding periodic effects and thus can easily learn to estimate the future trend conditioned on the lookback signals within a time window. In other words, we handle trend modeling in an auto-regressive style, as most other deep learning forecasting models did.
>
> In effect, the most challenging part lies in the initialization of periodic coefficients (will be discussed in depth in the response to your second question) when a global trend exists. Specifically, we develop a two-stage optimization procedure to initialize those periodic coefficients that not only fit historical observations well but also have good generalization capabilities. Therefore, even there is an obvious trend (e.g. upward trend) coupled with various periodic oscillations, which is often the case in Electricity and Traffic datasets, we can still identify those inherent periodic patterns  steadily and combine them with local momenta to foster forecasting.
>
> Question (2): “My another concern is with the training of the periodic module. If the periodic module is randomly initialized, can this DEPTS model still perform well? Would it still can accurately capture the obvious periods? I am very curious about the forecasting results without initialization of the periodicity module.”
>
> If the periodic module is randomly initialized and trained together with the expansion module from scratch via stochastic gradient descent (SGD), the periodic coefficients can hardly converge to a good solution with our desired properties of effective periodicity modeling. We have briefly mentioned this challenge at the end of the first paragraph in Section 4.3 of the paper. Here we would like to provide more insights with corresponding evidences to reveal the necessity of our customized initialization.
>
> First, as you have mentioned, the periodicity module considers a time series as a whole and thus captures certain global information. In contrast, the optimization of neural networks via SGD only takes a batch of time-series segments per each parameter update. Therefore, optimizing periodic coefficients from random initialization via SGD easily leads to unstable behaviors and usually takes massive gradient-updating steps to converge.
>
> Second, when stacking the expansion module over the periodic module, we introduce much more complexities into the optimization procedure. In such a highly non-convex condition, the SGD-based optimization for periodic coefficients $\phi$ becomes much more inefficient and is easily trapped into numerous local optima, which do not characterize our desired periodic properties.
>
> Moreover, we supplement additional experiments to provide concrete empirical evidence. Results are shown in the following table, where ‘RandInit’ denotes training DEPTS from random initialization. These empirical results also clearly demonstrate the importance of our initialization.
>
> | Model |   Electricity (2014-12-15) |   Traffic (2009-03-24) |   CAISO (2020-10-01) |   NP (2020-10-01) | M4 (Hourly) |
> | :------ | ------: | ------: | ------: | ------: | ------: |
> |N-BEATS | 0.17100 | 0.11200 | 0.02553 | 0.21130| 0.02300 |
> | RandInit     |    0.17463 |   0.11048 | 0.02496 | 0.20869 | 0.02401|
> | DEPTS   |    0.13915 |   0.10653 | 0.02008 | 0.17885 | 0.02050 |
>
> Finally, we further shed some light on how our periodic initialization helps the optimization procedure to promote better understanding. In this paper, we formulate a two-stage optimization problem to identify proper periodic coefficients, which describe the inherent periodicity (good generalization). Tackling this optimization is also nontrivial, thus we develop an approximation method based on the DCT tool from signal processing. This approximation procedure is conducted on the whole series for training instead of on a batch of series segments iteratively, thus it not only runs fast but also produces more stable estimates for the inherent periodicity. In this way, we find a good starting point for those periodic coefficients, which further facilitates the subsequent end-to-end optimization to produce our desired properties: 1) capturing the global periodicity and 2) modeling the complicated periodic dependencies.

---

> > ### Comment · Reviewer_3Zt1 · 2021-11-22
> > **Thank you for the detailed response**
> >
> > Thank you for the comprehensive responses to my questions. I appreciate it.

---

### Official Review · Reviewer_SZrb · 2021-11-03

**Correctness:** 4
**Technical Novelty And Significance:** 3
**Empirical Novelty And Significance:** 3
**Recommendation:** 6
**Confidence:** 3

**Main Review:**

In general, this paper is well-written and easy to follow. Periodicity is a property rarely explicitly handled by previous deep learning models. The proposed model achieves better performance compared to previous methods such as N-BEATS. Experiments on different settings and parameters are provided.

However, I have the following questions:
(1). Why does the expansion module need periodic blocks? Considering estimated hidden state z_t has already included inherent periodicity (assumed by the paper), why not directly taking x_t minus z_t to the local block for the predictions?

(2). The authors discuss the connections between the model and N-BEATS. What is unclear is the main difference between N-BEATS (interpretable) and DEPTS, since both of them have applied the periodicity. More illustrations should be provided.

(3). Other time series analysis baselines such as ARIMA should be included in the experiments, although they may not beat the proposed method.

**Summary Of The Paper:**

This paper proposes a deep learning model tailor-designed for periodic time series forecasting problem. The inner architecture is composed of an expansion module to tackle complicated dependencies between inherent periods and time series signals and a periodicity module to capture sophisticated period signals.

**Summary Of The Review:**

Overall the paper appears to have good quality, with a new time series prediction model proposed.

---

> ### Author Response · Authors · 2021-11-16
> **Response (Part 1/2)**
>
> Thanks very much for your review. We hope the answers below can help to address your questions properly.
>
> Question (1): “Why does the expansion module need periodic blocks? Considering estimated hidden state z_t has already included inherent periodicity (assumed by the paper), why not directly taking x_t minus z_t to the local block for the predictions?”
>
> Indeed, $z_t$ is a hidden state that characterizes the inherent periodicity, but periodic blocks are indispensable for DEPTS to produce more accurate PTS forecasting due to the following two reasons:
> 1.	The dependency of $x_t$ on $z_t$ can be very complicated in real-world scenarios. Simply feeding ($x_t – z_t$) into the local block implies a hard assumption of the additive seasonality, which restrains the modeling capabilities.
> 2.	Since ($x_t – z_t$) denotes the raw time-series signal subtracting the periodic effect, it is challenging for the model to forecast the future signals, $x_{t:t+H}$, solely based on the periodicity-agnostic inputs, ($x_{t-L:t} – z_{t-L:t}$). In contrast, our periodic blocks produce extra forecasts based on $z_{t-L:t+H}$.
>
> Accordingly, we develop periodic blocks to facilitate the layer-by-layer expansions of uncertain periodic dependencies and to produce a part of forecasts much related to periodic effects.  These considerations correspond to the motivations behind the residual equations (4) in the paper, and our ablation experiments in the appendix demonstrate the necessities of these residual connections.
>
> To provide more intuitive and convincing evidence for the above justifications, we further conduct additional ablation studies as you suggested. The following table illustrates the comparison results, where NoPeriodBlock denotes the ablated architecture that removes all periodic blocks by feeding ($x_t – z_t$) directly into the local block. We can observe that DEPTS consistently and significantly outperforms this variant. Moreover, you may check our responses to reviewer 3Zt1 and reviewer xtYj about other ablation studies on the periodicity module.
>
> | Model |   Electricity (2014-12-15) |   Traffic (2009-03-24) |   CAISO (2020-10-01) |   NP (2020-10-01) | M4 (Hourly) |
> | :------ | ------: | ------: | ------: | ------: | ------: |
> | N-BEATS | 0.17100 | 0.11200 | 0.02553 | 0.21130| 0.02300|
> | NoPeriodBlock   |    0.20615 |   0.11829 | 0.07275 | 0.22503 |0.03848|
> | DEPTS   |    0.13915 |   0.10653 | 0.02008 | 0.17885 | 0.02050|
>
>
> Question (2): “The authors discuss the connections between the model and N-BEATS. What is unclear is the main difference between N-BEATS (interpretable) and DEPTS, since both of them have applied the periodicity. More illustrations should be provided.”
>
> We really appreciate this suggestion . The main difference between N-BEATS and DEPTS lies in their variant capabilities in 1) capturing diversified periodicity in practice and 2) modeling global periodicity rather than temporary patterns merely within the lookback window. To elaborate on these two points, we provide more comparison discussions about DEPTS and N-BEATS from the interpretability aspect, as shown in the following paragraphs. We will merge them into the revised paper.
>
> N-BEATS did not model the global periodicity explicitly. Instead, it leveraged parameterized blocks to analyze the input signals and then produce proper coefficients for a set of basis vectors. The interpretability of N-BEATS lies in the specific designs of these basis vectors. For example, its seasonal basis vectors are [1, cos(2\pi t), …, cos(2\pi (H/2-1) t), sin(2\pi t), …, sin(2\pi (H/2-1) t)], where H denotes the length of forecasting horizon. In this way, N-BEATS can produce forecasts with periodic effects. However, compared with the global periodicity modeling of DEPTS, the seasonal modeling of N-BEATS has two limitations:
> 1.	The available period frequencies have been restricted to [1, …, (H/2-1)], which are insufficient to capture diversified periodicity in practice.
> 2.	The coefficients of these basis vectors are conditioned on the input signals within the lookback window. Therefore, the composition of these periods varying with inputs cannot give us a comprehensive view of the overall periodic behaviors.
> In contrast, DEPTS aims to capture a unified composition of diversified periods, reflecting the inherent periodic patterns of a specific series.

---

> > ### Author Response · Authors · 2021-11-16
> > **Response (Part 2/2)**
> >
> > Question (3): “Other time series analysis baselines such as ARIMA should be included in the experiments, although they may not beat the proposed method.”
> >
> > Thanks for this comment. The PARMA baseline included in current experiments is exactly a seasonal ARIMA model. We should make clearer explanations of this in the experimental setup and will address this issue in the revised paper.
> >
> > To be more specific, we use the AutoARIMA implementation provided by the sktime package (https://www.sktime.org/) to search for the best configurations of PARMA and then produce forecasts for evaluation. We have reported the performance of PARMA on CAISO and NP benchmarks. As for Electricity, Traffic, and M4 (Hourly) datasets, we include available SOTA models for a fair comparison, just as previous studies did. While for the synthetic data, AutoARIMA performs much worser than N-BEATS. Including its results interferes the readability of Figure 3 to compare N-BEATS with DEPTS. So, we only retain N-BEATS for clearer visualization.

---

### Author Response · Authors · 2021-11-17
**General Update**

Dear all reviewers and ACs,

Thanks very much for your review. We really appreciate your insightful questions and constructive suggestions. In addition to the respective responses, we have prepared a revised paper that attempts to address all the concerns and to adopt these valuable suggestions completely.

Following the suggestions from Reviewer SZrb, we add ablation experiments about removing periodic blocks, discuss the differences between N-BEATS and DEPTS  on periodicity modeling, and clarify the description on the PARMA baseline.

Following the suggestions from Reviewer 3Zt1, we supplement the experiments to compare the random initialization of periodic coefficients with our customized initialization.

Following the suggestions from Reviewer xtYj, we further add comparison experiments, which includes fixing the periodicity module after initialization and treating the periodic states as a covariate of the raw signals, to verify the advantages of our expansion learning.

Following the suggestions from Reviewer qo99, we attempt to improve the writing of Section 4.2 by providing the motivations before delving into detailed description. Besides, we supplement the complexity analyses of our approximation algorithm to the appendix.

Please let us know if you have extra questions.

Thanks,

All authors

---

### Decision · Program_Chairs · 2022-01-20

**Decision:**

Accept (Spotlight)

**Comment:**

The paper proposed a novel deep learning model specifically designed for periodic time series forecasting problems. The approach includes lay-by-layer expansion, residual learning, and periodic parametrization. The model outperforms state-of-the-art baselines on several time series forecasting benchmarks.  The reviewers appreciate the extensive experimental results, but also suggested improvement on writing and comparison regarding the parameter efficiency of the model.